# Latent Diffusion Model without Variational Autoencoder

**Minglei Shi**[1][*]   **Haolin Wang**[1][*]   **Wenzhao Zheng**[1][†]   **Ziyang Yuan**[2]   **Xiaoshi Wu**[2]
**Xintao Wang**[2]   **Pengfei Wan**[2]   **Jie Zhou**[1]   **Jiwen Lu**[1][‡]
[1]Department of Automation, Tsinghua University   [2]Kling Team, Kuaishou Technology

**Project Page:**   https://howlin-wang.github.io/svg
**Code Repository:**   https://github.com/shiml20/SVG

## Abstract

Recent progress in diffusion-based visual generation has largely relied on latent diffusion models with Variational Autoencoders (VAEs). While effective for high-fidelity synthesis, this VAE+Diffusion paradigm still suffers from limited training and inference efficiency, along with poor transferability to broader vision tasks. These issues stem from a key limitation of VAE latent spaces: the lack of clear semantic separation and strong discriminative structure. Our analysis confirms that these properties are not only crucial for perception and understanding tasks, but also equally essential for the stable and efficient training of latent diffusion models. Motivated by this insight, we introduce **SVG**—a novel latent diffusion model without variational autoencoders, which unleashes **S**elf-supervised representations for **V**isual **G**eneration. SVG constructs a feature space with clear semantic discriminability by leveraging frozen DINO features, while a lightweight residual branch captures fine-grained details for high-fidelity reconstruction. Diffusion models are trained directly on this semantically structured latent space to facilitate more efficient learning. As a result, SVG enables accelerated diffusion training, supports few-step sampling, and improves generative quality. Experimental results further show that SVG preserves the semantic and discriminative capabilities of the underlying self-supervised representations, providing a principled pathway toward task-general, high-quality visual representations.

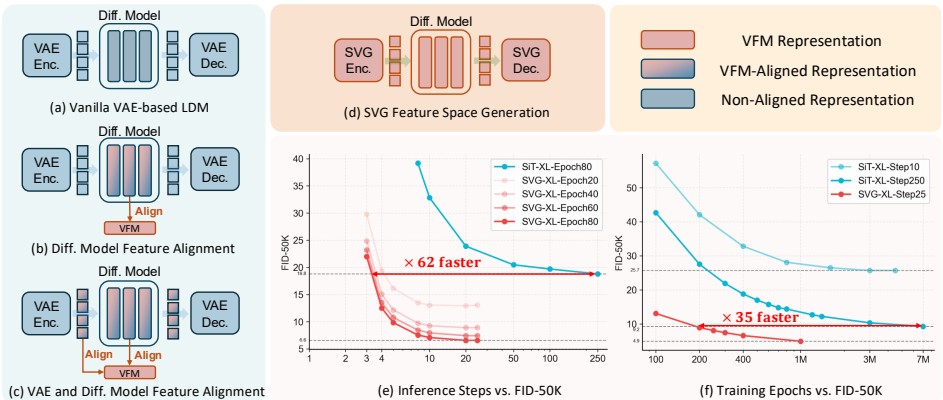

Figure 1: **Core contribution of SVG.** (a) Vanilla VAE-based LDM: the diffusion model is trained on the pretrained VAE latent space. (b) Diff. Model Feature Alignment: intermediate features of the diffusion model are aligned to Visual Foundation Model (VFM) features. (c) VAE and Diff. Model Feature Alignment: both VAE latent features and diffusion model intermediate features are aligned to VFM features. (d) Our method: the diffusion model is trained directly in the SVG space derived from self-supervised representations (DINOv3). (e-f) Comparisons of inference and training efficiency.

---

[*]Equal contribution. Listed in alphabetical order.

[†]Project leader.

[‡]Corresponding author.

# 1 INTRODUCTION

Generative models have made remarkable progress in recent years, with diffusion models (Rombach et al., 2021; Ho et al., 2020; Song et al., 2021b; Liu et al., 2022; Lipman et al., 2023) emerging as a dominant paradigm. They have attracted substantial attention and demonstrated broad applicability across diverse scenarios, including text-to-image generation (Rombach et al., 2021; Chen et al., 2024; Esser et al., 2024; Labs, 2024), text-to-video generation (Yang et al., 2024; Wan et al., 2025; HaCohen et al., 2024), and beyond. Due to the inherently high-dimensional nature of visual data, training diffusion models directly at the pixel level remains challenging. To address this, mainstream approaches rely on pretrained variational autoencoders to compress raw visual data into a compact latent space, on which diffusion models are subsequently trained (Rombach et al., 2021).

Despite their success, the VAE+Diffusion paradigm exhibits several critical limitations. First, both training and inference are computationally expensive: for instance, training an ImageNet $256 \times 256$ generation model with the standard DiT implementation (Peebles & Xie, 2022) requires 7M steps, and inference typically demands more than 25 sampling steps to achieve satisfactory results. As depicted in parts (a)-(c) of Figure 1, although some recent methods (Yu et al., 2025; Leng et al., 2025; Yao et al., 2025) attempt to accelerate diffusion training by aligning with external feature spaces of vision foundation models (VFM) (Oquab et al., 2023; He et al., 2021; Chen et al., 2020d; Radford et al., 2021; Zhai et al., 2023) or imposing regularization constraints on the VAE latent space (Wang & He, 2025; Stoica et al., 2025), these approaches provide only ad hoc fixes, as they do not fundamentally alter the semantic structure of the latent space, which remain weakly discriminable as clearly shown in Figure 4a. Importantly, VAE latent representations are generally not employed in modern multi-modal large models, and their restricted perceptual capabilities (Yin et al., 2024; Jin et al., 2024) highlight a fundamental limitation. This discrepancy implies that VAE latents are unlikely to serve effectively as unified visual representations.

In this paper, we argue that a discriminative semantic structure in the latent space can substantially facilitate the training of diffusion models. By leveraging the powerful self-supervised features from DINOv3 (Siméoni et al., 2025), we demonstrate that it is possible to construct a feature space that enables efficient diffusion training in a simple yet effective way, while fully retaining DINOv3's strengths beyond generation.

We start by analyzing the semantic distributions of various VAE latent spaces to examine the limitations of the conventional VAE+Diffusion paradigm. Our study indicates that semantic entanglement in the vanilla VAE latents is a major obstacle to efficient diffusion. This observation leads to two key insights: first, VAE latents may not be optimal for latent diffusion models; second, since visual perception and understanding tasks also benefit from semantically structured representations, it is feasible to design a single unified feature space that simultaneously supports all core vision tasks.

Specifically, we examine several state-of-the-art visual representations in terms of image reconstruction, perception, and semantic understanding. We find that DINOv3 features offer the greatest potential as a unified feature space, as they preserve substantial coarse-grained image information and inherently exhibit strong semantic discriminability. To further enhance generation quality, we augment the frozen DINOv3 encoder with a lightweight Residual Encoder that captures the missing fine-grained perceptual details. The residual outputs are concatenated with the DINOv3 features along the channel dimension to enrich the representation, and subsequently aligned with the original DINOv3 feature distribution to preserve semantic structure. The resulting SVG feature space combines strong semantic discriminability with rich perceptual detail, leading to more efficient training of diffusion models, improved generative quality, and enhanced inference efficiency.

We highlight the following significant contributions of this paper:

- We systematically analyze the limitations of mainstream VAE latent spaces in latent diffusion models, highlighting how semantic dispersion may affect the efficiency of generative modeling.

- We propose SVG, a latent diffusion model without variational autoencoders, built upon a unified feature space that retains the potential to support multiple core vision tasks beyond generation.

- SVG Diffusion achieves impressive generative quality while ensuring rapid training and highly efficient inference.

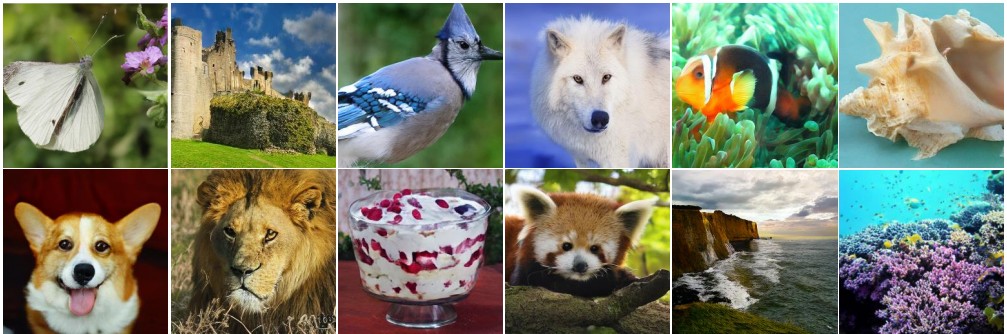

Figure 2: **Selected 256×256 samples from SVG-XL**. We use a cfg of 4.0 and 25 Euler steps.

## 2 RELATED WORKS

**Visual generation.** Generative models aim to learn the underlying probability distribution of data and to generate novel samples that are both realistic and diverse. Generative adversarial networks (GANs) (Goodfellow et al., 2014; Radford et al., 2016; Arjovsky et al., 2017; Gulrajani et al., 2017; Karras et al., 2019; Zhu et al., 2017; Karras et al., 2018; Sauer et al., 2022) generate realistic images via adversarial training but often suffer from mode collapse, instability, and poor interpretability. Another family of approaches follows an autoregressive paradigm, where an image is represented as a sequence of pixels, patches, or latent tokens. The joint distribution is factorized into conditional probabilities and modeled sequentially, as in (Salimans et al., 2017; Vaswani et al., 2018; Chen et al., 2020a). Extensions based on masked autoregression (He et al., 2021; Chang et al., 2022; Li et al., 2024) predict missing tokens given visible context, analogous to masked language models in NLP. This formulation enables direct transfer of transformer-based sequence modeling techniques to large-scale image generation. More recently, diffusion models (Ho et al., 2020; Nichol & Dhariwal, 2021; Song et al., 2021a;b) have emerged as a powerful alternative, generating images by iteratively denoising Gaussian noise. They achieve state-of-the-art fidelity and diversity, with improved training stability and mode coverage. An improved version, the latent diffusion model (LDM) (Rombach et al., 2021; Peebles & Xie, 2022; Ma et al., 2024; Liu et al., 2022), integrates a VAE (Kingma & Welling, 2022) with the diffusion process to operate in a lower-dimensional latent space, reducing computational cost while maintaining generation quality. Nevertheless, these models still require multiple inference steps, limiting generation speed. A line of research aims to improve the structural properties of the VAE latent space (Burgess et al., 2018; Xu & Durrett, 2018; Kouzelis et al., 2025a; Skorokhodov et al., 2025) or to incorporate diffusion processes into VAE reconstruction (Preechakul et al., 2022; Pandey et al., 2021; Vahdat et al., 2021). More recently, efforts have focused on aligning the latent features of diffusion models with external semantic representations (Yu et al., 2025; Leng et al., 2025; Yao et al., 2025; Li et al., 2023b), while other works explore joint generation approaches that integrate discriminative components into the generative process (Wu et al., 2025; Kouzelis et al., 2025b). Despite these advances, we observe that even with structural improvements to the VAE latent space or alignment with external semantic features, the semantic discriminability of VAE representations remains inherently limited. Consequently, the VAE+Diffusion paradigm is constrained in both training and inference efficiency, and it fails to establish a unified feature space capable of supporting visual generation, perception, and understanding.

**Visual representation learning.** Recent advances in visual representation learning can be broadly categorized into discriminative, generative, and multimodal paradigms. Discriminative methods, such as self-supervised learning (SSL) approaches including DINO (Zhang et al., 2022; Oquab et al., 2023; Siméoni et al., 2025), SimCLR (Chen et al., 2020b;c), MoCo (He et al., 2019; Chen et al., 2020d), and BYOL (Grill et al., 2020), learn informative features without explicitly modeling the data distribution, producing highly separable representations that excel in classification, retrieval, and dense prediction, but often discard fine grained generative information, limiting their utility for synthesis or reconstruction. Generative approaches, including VAE (Kingma & Welling, 2022), masked autoencoders (MAE) (He et al., 2021), masked image modeling (MIM) (Xie et al., 2022b), and diffusion models (Ho et al., 2020; Song et al., 2021b), capture rich contextual and perceptual information by reconstructing inputs or modeling the underlying data distribution, providing embeddings beneficial for downstream tasks; however, they are computationally intensive and may produce representations that are less discriminative than contrastive methods. Multimodal methods, exemplified by CLIP (Radford et al., 2021), SigLIP (Zhai et al., 2023; Tschannen et al., 2025), Florence (Xiao et al., 2023), and BLIP (Li et al., 2022; 2023a), align images and text in a shared latent

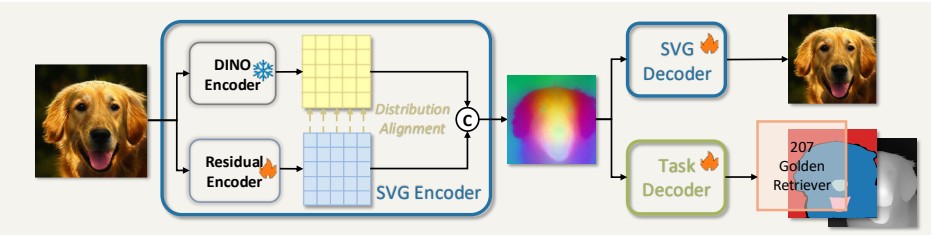

Figure 3: **Architecture of the proposed SVG Autoencoder.** The model augments the DINO encoder with a Residual Encoder to achieve high-quality reconstruction and preserve transferability.

space, enabling zero-shot learning, cross-modal retrieval, and enhanced semantic understanding, but rely on large-scale paired data and can underperform when one modality dominates or data quality is uneven. Despite these advances, existing visual representations often struggle to provide a unified solution for various visual tasks. Here, we for the first time demonstrate that features learned via self-supervised methods can be directly repurposed for generative modeling, enabling the construction of a unified feature space that effectively supports diverse core vision tasks.

## 3 METHODOLOGY

### 3.1 PRELIMINARIES

**Diffusion models.** Diffusion Models (Ho et al., 2020; Rombach et al., 2021; Song et al., 2021b) have been the dominant generative modeling for continuous feature space, which can transform the Gaussian distribution to the data distribution through iterative inference. The diffusion process can be represented as follows:

$$\mathbf{x}_t = \alpha_t \mathbf{x}_0 + \sigma_t \epsilon, \quad t \in [0, 1], \quad \epsilon \sim \mathcal{N}(0, \mathbf{I}). \tag{1}$$

where $\alpha_t$ and $\sigma_t$ are monotonically decreasing and increasing function of $t$, respectively. And the marginal distribution $p_1(\mathbf{x})$ converges to $\mathcal{N}(0, \mathbf{I})$, when $\alpha_1 = 0, \sigma_1 = 1$, $p_0(\mathbf{x})$ converges to data distribution, when $\alpha_0 = 1, \sigma_0 = 0$. We train the model using a denoising loss as follows:

$$\mathcal{L}_{\text{DDPM}} = \mathbb{E}_{\mathbf{x}_0 \sim p_0(\mathbf{x}), \epsilon \sim p_1(\mathbf{x})}[\lambda(t)\|\epsilon_\theta(\mathbf{x}_t, t) - \epsilon_t\|]. \tag{2}$$

where $\lambda_t$ is a time-dependent coefficient and $\epsilon_t$ is the Gaussian noise added to $\mathbf{x}_t$. And sampling from a diffusion model can be achieved by solving the reverse-time SDE or the corresponding diffusion ODE (Song et al., 2021b).

Recently, flow-based generative models (Liu et al., 2022; Lipman et al., 2023; Esser et al., 2024) have emerged as a leading approach for generative modeling using flow matching. These methods construct a velocity field that interpolates between a Gaussian distribution and the data distribution:

$$\mathbf{x}_t = (1 - t)\mathbf{x}_0 + t\epsilon, \quad t \in [0, 1], \quad \epsilon \sim \mathcal{N}(0, \mathbf{I}), \tag{3}$$

$$\mathbf{v}_t \triangleq \frac{\mathrm{d}\mathbf{x}_t}{\mathrm{d}t} = \epsilon - \mathbf{x}_0. \tag{4}$$

The flow matching objective is then formulated as

$$\mathcal{L}_{\text{FM}} = \mathbb{E}_{\mathbf{x}_0 \sim p_0(\mathbf{x}), \epsilon \sim p_1(\mathbf{x})}[\lambda(t)\|\mathbf{v}_\theta(\mathbf{x}_t, t) - \mathbf{v}_t\|]. \tag{5}$$

Sampling from a flow-based model can be achieved by solving the probability flow ODE.

### 3.2 RETHINKING LATENT DIFFUSION MODELS

Latent diffusion models (Rombach et al., 2021) trained on VAE latent space have emerged as the leading paradigm for visual generation and have been widely adopted in advanced diffusion frameworks. By compressing images into a lower-dimensional latent space, these models focus on learning essential semantic structures while ignoring imperceptible high-frequency details, effectively separating perceptual compression from semantic generation (Rombach et al., 2021). However, training in VAE latent spaces remains time- and resource-intensive, and controlling the degree of

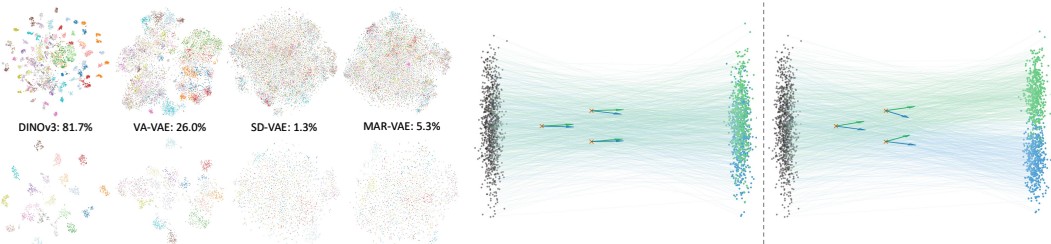

(a) The t-SNE visualization of different visual feature spaces.

(b) Toy example illustrating the impact of semantic dispersion in the feature space on diffusion model training.

Figure 4: **Visualization of feature space.** (a) Feature visualization with t-SNE for 100 ImageNet classes (100 random samples per class, top row) and 20 classes (100 random samples per class, bottom row). Features are extracted using DINOv3 (Siméoni et al., 2025), VA-VAE (Yao et al., 2025), SD-VAE (Rombach et al., 2021), and MAR-VAE (Li et al., 2024), with each class shown in a distinct color, and model names annotated with their linear-probe Top-1 accuracy on ImageNet-1K (Deng et al., 2009). (b) Each subfigure shows the source distribution (black dots) and the target distribution, where the samples are divided into two semantic categories (green and blue dots). The arrows indicate the directions of the mean velocity field at each point.

perceptual compression is challenging, leading to the common dilemma that better reconstruction often results in worse generation (Esser et al., 2024; Gupta et al., 2025; Kilian et al., 2024; Yao et al., 2025). Recent studies show that aligning either diffusion model hidden states or VAE latents with VFM features can substantially accelerate training (Yu et al., 2025; Leng et al., 2025; Yao et al., 2025), prompting the question of which VFM properties are critical for this improvement.

To investigate this, we perform t-SNE visualizations of commonly used VAE latent spaces, as depicted in Figure 4a. Specifically, VA-VAE (Yao et al., 2025) aligns VAE latents with DINO (Oquab et al., 2023) features. We observe that vanilla VAE latents exhibit strong semantic entanglement: representations from different classes are heavily mixed. After alignment with a VFM, inter-class separation increases, while intra-class representations become more compact.

We further illustrate this effect with a toy example in Figure 4b. When the latent space exhibits clear separation between semantic classes (right), the mean velocity directions are consistent within each class—latents from the same class move in similar directions—and distinct across classes, with different classes showing clearly divergent directions at the same point. Such structured dynamics simplify optimization, allowing high-quality results to be achieved with fewer sampling steps. In contrast, when the latent space is highly entangled (left), velocity directions from different classes overlap and become ambiguous, complicating training and requiring more sampling steps.

These findings underscore the importance of semantic dispersion for latent diffusion model training. The conventional reliance on VAE latents arises from the fact that semantic features alone are inadequate for high-fidelity reconstruction. Nevertheless, our results demonstrate that with modern VFMs, one can construct a general-purpose latent space that simultaneously provides discriminative semantic structure and robust reconstruction capability.

### 3.3 VISUAL FEATURE GENERATION

Based on the analysis in Section 3.2, we propose SVG, a novel generative paradigm that constructs a task-general feature space combining the semantic discriminability of vision foundation models with the fine-grained perceptual details required for high-quality generation. The overall architecture of SVG is shown in Figure 3.

**SVG autoencoder.** The SVG autoencoder is designed to preserve the semantic structure of frozen DINO features while supplementing them with residual perceptual information that is crucial for faithful image reconstruction. Concretely, it consists of two components: a frozen DINOv3 encoder and a lightweight Residual Encoder built on a Vision Transformer (Dosovitskiy et al., 2021). The Residual Encoder captures fine-grained details that are missing in DINO features, and its outputs are concatenated along the channel dimension with the DINO features to form the complete SVG feature. The SVG Decoder, following the VAE decoder design from (Rombach et al., 2021), maps the SVG feature back to pixel space. This architecture is intentionally simple and lightweight, avoiding complex

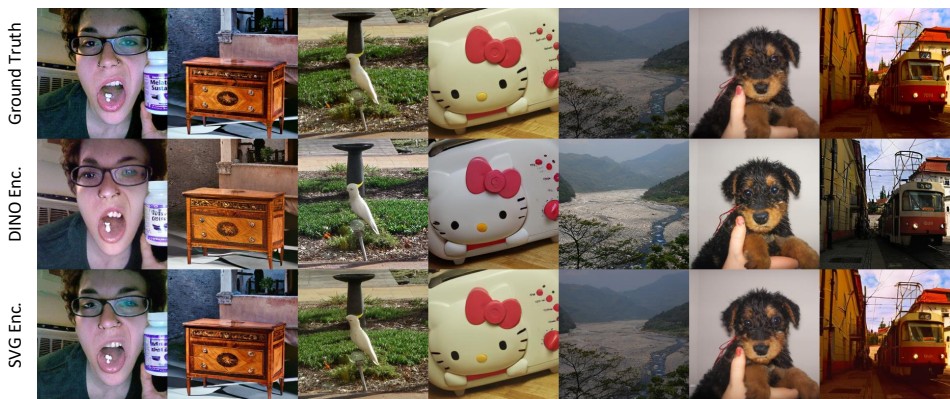

Figure 5: **Visualization of SVG reconstruction.** Incorporating the Residual Encoder enables SVG to better preserve visual information, such as color and high-frequency details.

modifications while achieving the dual goals of retaining DINOv3's strong semantic discriminability and enhancing it with detailed perceptual information. The importance of the Residual Encoder is further illustrated in Figure 5, which shows that omitting it noticeably reduces reconstruction quality, particularly for color and fine-grained details.

**SVG diffusion.** Unlike prior approaches that construct diffusion models on low-dimensional VAE latent spaces (Rombach et al., 2021), SVG Diffusion treats the high-dimensional SVG feature space as the target distribution, trained using the flow matching objective defined in Equation (5). Specifically, for $256 \times 256$ images, the DINOv3-ViT-S/16+ encoder produces a $16 \times 16 \times 384$ feature map, compared with the $16 \times 16 \times 4$ VAE latent in DiT (Peebles & Xie, 2022). While training diffusion models in such high-dimensional spaces is generally challenging and prone to unstable convergence (Xie et al., 2024), the well-dispersed semantic structure of SVG features makes training stable and efficient. Consequently, SVG Diffusion converges faster and achieves superior generative quality compared with VAE-based diffusion. Moreover, since hidden states in diffusion models typically have channel sizes larger than 384 (Peebles & Xie, 2022), SVG Diffusion does not incur inference inefficiency. As shown in Section 4.3, its strong semantic continuity also enables few-step sampling, yielding superior inference efficiency.

**SVG training pipeline.** The training is conducted in two stages. In the first stage, we optimize only the Residual Encoder and the SVG Decoder with the reconstruction loss defined in (Rombach et al., 2021). However, naively training in this way causes the decoder to over-rely on the Residual Encoder, and the mismatch in numerical ranges between the DINO and residual outputs can compromise the semantic discriminability inherited from DINO. To address this, we align the Residual Encoder outputs with the DINO feature distribution, ensuring that the added residual dimensions do not distort the original semantic space. In specific, for each training batch, let $F_D$ denote the DINOv3 features and $F_R$ the residual features. The alignment is performed by normalizing $F_R$ to match the batch statistics of $F_D$:

$$\hat{F}_R = \frac{F_R - \mu(F_R)}{\sigma(F_R)} \cdot \sigma(F_D) + \mu(F_D) \tag{6}$$

where $\mu(\cdot)$ and $\sigma(\cdot)$ compute the mean and standard deviation along the feature dimension.

In the second stage, SVG Diffusion is trained under the settings of SiT (Ma et al., 2024), with QK-Norm (Henry et al., 2020) applied and the per-channel SVG feature space normalized to stabilize training.

## 4 EXPERIMENTS

In this section, we validate the feasibility and effectiveness of the proposed SVG through extensive experiments. Specifically, we investigate the following key questions:

- Can SVG, as a latent diffusion model without VAE, achieve competitive generative quality, high training efficiency (Table 1), fast inference (Table 2), and favorable scaling properties (Table 2)?

Table 1: **System-level performance on ImageNet** $256 \times 256$ **for SVG.** Operating in a *unified feature space*, SVG achieves high-quality *few-step generation* (25 steps), surpassing baseline models and converging faster. [†] indicates reproduction results; for flow-matching, only the ODE solver is used.

| Method (SVG) | Reconstruction / Tokenizer | | Training Epochs | Steps | #params | Generation w/o CFG | | Generation w/ CFG | |
|---|---|---|---|---|---|---|---|---|---|
| | Tokenizer | rFID | | | | gFID | IS | gFID | IS |
| *Generation-Specific Feature Space* | | | | | | | | | |
| LlamaGen (Sun et al., 2024) | VQGAN | 0.59 | 300 | 256 | 3.1B | 9.38 | 112.9 | 2.18 | 263.3 |
| MaskDiT-XL (Zheng et al., 2024) | SD-VAE | 0.61 | 1600 | 250 | 675M | 5.69 | 177.9 | 2.28 | 276.6 |
| DiT-XL (Peebles & Xie, 2022) | SD-VAE | 0.61 | 1400 | 250 | 675M | 9.62 | 121.5 | 2.27 | 278.2 |
| SiT-XL (Ma et al., 2024) | SD-VAE | 0.61 | 1400 | 250 | 675M | 9.35 | 126.6 | 2.15 | 258.1 |
| REPA-XL (Yu et al., 2025) | SD-VAE | 0.61 | 800 | 250 | 675M | 5.90 | - | 1.42 | 305.7 |
| REPA-XL (Yu et al., 2025) | SD-VAE | 0.61 | 80 | 250 | 675M | 7.90 | 118.6 | 2.65 | 268.62 |
| SiT-XL[†] | VA-VAE | 0.26 | 80 | 250 | 675M | 5.96 | 128.0 | 3.63 | 290.6 |
| *Few-Step Generation* | | | | | | | | | |
| SiT-XL[†] | SD-VAE | 0.61 | 80 | 25 | 675M | 22.58 | 67.3 | 6.06 | 169.5 |
| SiT-XL[†] | VA-VAE | 0.26 | 80 | 25 | 675M | 7.29 | 121.0 | 4.13 | 279.7 |
| *Task-General Feature Space* | | | | | | | | | |
| SVG-XL | SVGTok | 0.65 | 80 | 25 | 675M | 6.57 | 137.9 | 3.54 | 207.6 |
| SVG-XL | SVGTok | 0.65 | 500 | 25 | 675M | 3.94 | 169.3 | 2.10 | 258.7 |
| SVG-XL | SVGTok | 0.65 | 1400 | 25 | 675M | 3.36 | 181.2 | 1.92 | 264.9 |

- Does the SVG feature space provide task-general representations applicable across diverse vision tasks (Table 4, Figure 6)?
- Are the choices of VFMs (Table 3) and the components of the SVG Encoder (Table 4) reasonable?

## 4.1 EXPERIMENT SETUPS

**Training details.** All the models were trained on ImageNet1K (Russakovsky et al., 2015) dataset. For the reconstruction task, we follow the settings of VA-VAE (Yao et al., 2025) and employ the same decoder architecture. The additional encoder is implemented as a Vision Transformer using the `timm` library (Wightman, 2019). We jointly train the residual encoder and SVG decoder. For visual generation, we strictly follow the training setups in SiT (Ma et al., 2024). To ensure a fair comparison, we keep the main architecture unchanged and only replace the patch embedding layer with a simple linear projection that maps the feature dimension to the model dimension.

**Metrics.** We adopt reconstruction FID (rFID)(Heusel et al., 2017), PSNR, LPIPS(Zhang et al., 2018), and SSIM (Wang et al., 2004) to evaluate reconstruction quality. For image generation, we report FID (gFID)(Heusel et al., 2017) and Inception Score (IS)(Salimans et al., 2016), providing results both with and without classifier-free guidance (CFG).

## 4.2 MAIN RESULTS

We evaluate the system-level performance of SVG on ImageNet $256 \times 256$, comparing against representative baselines. In generation-specific feature spaces, baselines typically require 64–256 steps to produce high-quality samples, but their performance drops sharply under few-step generation (25 steps); for example, SiT-XL[†] attains a gFID of 22.58 without classifier-free guidance. In contrast, SVG-XL, operating in a task-general feature space with the proposed SVG Autoencoder, delivers consistently superior results. The reconstruction metric (rFID=0.65) confirms strong fidelity. Under 25-step generation with 80 training epochs, SVG-XL achieves gFID=6.57 (w/o CFG), and gFID=3.54 (w/ CFG), substantially outperforming all baseline models. With extended training for 500 epochs, performance further improves to gFID=3.94 (w/o CFG) and gFID=2.10 (w/ CFG), competitive with generation-specialized SOTA methods while simultaneously supporting multiple tasks. These results highlight that the unified SVG feature space enables faster diffusion model training, efficient few-step inference, and high-quality image generation.

## 4.3 ANALYSIS

**Preserving the original capabilities of DINO features.** The previous experiments have verified the superiority of the SVG space in visual generation. To test whether it also preserves visual perception and understanding capability, we evaluate the SVG encoder against the DINO encoder on downstream tasks where DINO is known to excel. For each task, we adopt a simple strategy: a lightweight MLP- or linear-layer decoder is appended to the encoder to map features into predictions, and only the decoder is trained (details in Appendix B). As reported in Table 4, the SVG feature maintains the

Table 2: **Comparison of few-Step generation and model scaling.** Both (a) and (b) report FID-50K results after 80 training epochs. (a) SVG achieves substantially better performance than SiT under few-step sampling. (b) SVG consistently outperforms SiT across different capacities with fewer sampling steps. SD and VA denote SD-VAE and VA-VAE, respectively.

**(a)** Few-step generation

| Method | Steps | FID-50K | |
|---|---|---|---|
| | | w/o CFG | w/ CFG |
| *Few-step generation* | | | |
| SiT-XL (SD) | 5 | 69.38 | 29.48 |
| SiT-XL (VA) | 5 | 74.46 | 35.94 |
| SVG-XL | 5 | 12.26 | 9.03 |
| SiT-XL (SD) | 10 | 32.81 | 10.26 |
| SiT-XL (VA) | 10 | 17.41 | 6.79 |
| SVG-XL | 10 | 9.39 | 6.49 |

**(b)** Model size scaling

| Method | #Params | Steps | FID-50K | |
|---|---|---|---|---|
| | | | w/o CFG | w/ CFG |
| *Model scaling* | | | | |
| SiT-B (SD) | 130M | 250 | 33.00 | 13.40 |
| SVG-B | 130M | 25 | 21.90 | 11.49 |
| SiT-L (SD) | 458M | 250 | 18.80 | 6.03 |
| SVG-L | 458M | 25 | 10.56 | 5.96 |
| SiT-XL (SD) | 675M | 250 | 17.20 | 5.10 |
| SiT-XL (VA) | 675M | 250 | 5.63 | 3.63 |
| SiT-XL (VA) | 675M | 25 | 7.29 | 4.13 |
| SVG-XL | 675M | 25 | 6.57 | 3.54 |

Table 3: **Comparison of different encoders and feature spaces.**. Reconstruction performance is reported after 5 epochs of training. (✔: advantage, ✔: partial, ✗: weak)

| | | Encoder Comparison | | | | Feature Space Comparison | | |
|---|---|---|---|---|---|---|---|---|
| Encoder | #params | Reconstruction Performance | | | | Semantic | Reconstruction | Perception |
| | | rFID↓ | PSNR↑ | LPIPS↓ | SSIM↑ | | | |
| SigLIP2 | 86M | 4.05 | 20.09 | 0.30 | 0.46 | ✔ | ✗ | ✗ |
| MAE | 86M | 1.69 | 25.04 | 0.18 | 0.69 | ✔ | ✔ | ✔ |
| DINOv2 | 22M | 2.18 | 18.10 | 0.30 | 0.40 | ✔ | ✔ | ✔ |
| DINOv3 | 29M | 1.87 | 18.44 | 0.31 | 0.41 | ✔ | ✔ | ✔ |
| SVG | 29M+11M | 1.60 | 21.77 | 0.25 | 0.55 | ✔ | ✔ | ✔ |

strong generalization ability of DINO, achieving comparable or even slightly superior results on ImageNet-1K (Deng et al., 2009; Russakovsky et al., 2015) classification, ADE20K (Zhou et al., 2019) semantic segmentation, and NYUv2 (Nathan Silberman & Fergus, 2012) depth estimation. Combined with its previously demonstrated strength in generative tasks, this dual advantage establishes the feature space produced by SVG Encoder as a unified representation space for diverse vision tasks.

**Effectiveness of SVG encoder.** We first compare the image reconstruction performance of several vision encoders. As shown in Table 3, SigLIP2 (Tschannen et al., 2025) exhibits poor reconstruction quality with high rFID scores. MAE (He et al., 2021), owing to its generative pretraining, achieves the best reconstruction results among the tested methods. The DINO series provides only limited reconstruction capabilities, while SVG enhances DINO with a Residual Encoder that captures fine-grained perceptual details, leading to substantially improved reconstruction quality. Considering both these results and prior studies, we observe that SigLIP2, which emphasizes global semantics while neglecting local details, performs poorly on reconstruction and perceptual tasks. MAE, despite its strong reconstruction ability, falls significantly behind DINO on semantic understanding and dense prediction tasks (Oquab et al., 2023). These findings indicate that neither SigLIP2 nor MAE is well-suited for constructing a unified feature space. In contrast, the SVG encoder retains DINO's strong semantic representation ability while achieving satisfactory reconstruction performance, making it an ideal basis for building a unified feature space.

To further assess the design of the SVG encoder, we conduct a detailed analysis of the Residual Encoder. As shown in Table 4, relying solely on DINOv3 features provides only limited reconstruction capability. Introducing a Residual Encoder markedly improves reconstruction. However, when residual features are naively concatenated with DINO features, the resulting feature distribution becomes imbalanced, disrupting the latent space's semantic dispersion. This degradation directly impacts generative performance, with gFID increasing from 6.12 to 9.03. Aligning the distribution of residual features with the frozen DINO features effectively addresses this issue, maintaining faithful reconstruction while facilitating the latent diffusion training. The above experimental results substantiate the effectiveness of the SVG design, demonstrating that its concise architecture is sufficient to ensure both faithful reconstruction and high-quality generation.

**Inference efficiency.** In Section 3.2, we noted that in latent spaces with high semantic dispersion and strong discriminability, the mean velocity directions of different semantic classes are more clearly separated. Furthermore, within each component, the velocity directions across spatial locations

Table 4: **Ablation study on the effectiveness of SVG encoder components.** Reconstruction performance is reported after 40 epochs of training, while generative metrics are evaluated after 500K training iterations using classifier-free guidance. For visual downstream tasks, we report fine-tuning results on ImageNet-1K, ADE20K, and NYUv2.

| Tokenizer | Reconstruction | | | | Generation | ImageNet-1K | | ADE20K | | NYUv2 | |
|---|---|---|---|---|---|---|---|---|---|---|---|
| | rFID↓ | PSNR↑ | LPIPS↓ | SSIM↑ | gFID↓ (w/ CFG) | Top-1↑ | Top-5↑ | mIoU↑ | mAcc↑ | RMSE↓ | A.Rel↓ |
| DINOv3 | 1.17 | 18.82 | 0.27 | 0.43 | 6.12 | 81.71 | 95.79 | 46.37 | 57.55 | 0.362 | 0.101 |
| +Residual Encoder | 0.78 | 24.25 | 0.19 | 0.67 | 9.03 | – | – | – | – | – | – |
| +Distribution Align. | 0.65 | 23.89 | 0.19 | 0.65 | 6.11 | 81.80 | 95.87 | 46.51 | 58.00 | 0.361 | 0.101 |

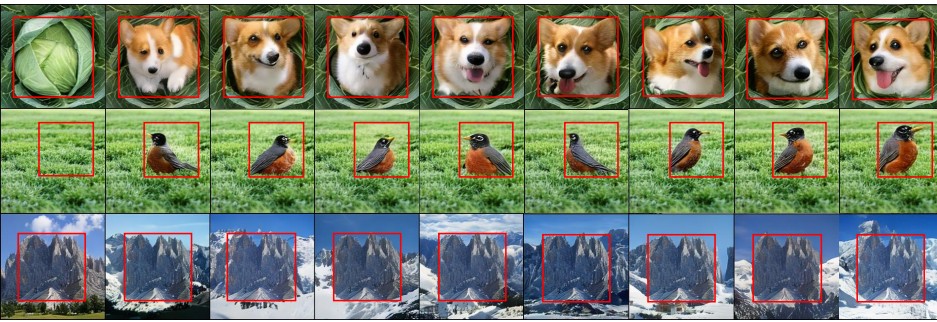

Figure 6: **Zero-shot class-conditioned editing using SVG.** The first column shows the original image. The first two rows edit the region inside the red box, whereas the third row edits the outside.

are more consistent. As a direct consequence, the discretization error during sampling is reduced, which in turn improves the quality of few-step sampling. The results in Table 2 clearly demonstrate this point. Under the same few-step sampling conditions (e.g., 5 or 10 steps), our method achieves significantly better performance than the baseline, both with and without CFG.

**Effect of model scaling** We next analyze how SVG behaves under different model capacities. As reported in Table 2, scaling up consistently improves the generative quality for both SiT and SVG, but SVG maintains a clear advantage at every scale. Notably, while SiT requires 250 steps to reach reasonable FIDs, SVG achieves substantially lower FIDs with only 10 steps. The relative improvements over SiT(SD) remain stable, indicating that the benefits of SVG do not diminish as model size increases. This confirms that the proposed feature space enables diffusion models to scale efficiently with model capacity.

**Zero-shot image editing.** To further assess SVG's generalization, we perform zero-shot class-conditioned editing. Following an SDEdit-style (Meng et al., 2021) procedure, input images are first inverted along the diffusion trajectory and selected regions replaced with noise. Sampling under the same class condition then generates the edits. As shown in Figure 6, SVG generates coherent edits that accurately follow the target class semantics while preserving consistency in non-edited regions. These results demonstrate that the SVG feature space exhibits strong semantic structure and inherent editability, enabling effective transfer to downstream generative tasks without the need for task-specific finetuning. Further experimental details are provided in Appendix C.

**Interpolation test.** To evaluate the continuity of SVG feature space, we perform latent space interpolation between two randomly sampled noise vectors conditioned on the same class embedding in Figure 7. We compare direct linear interpolation with spherical linear interpolation, which better preserves vector norms. In our experiments, SVG generates smooth, high-quality images under both interpolations, whereas VAE-based methods usually degrade under direct linear interpolation Figure 8. These results demonstrate that SVG feature space is continuous and robust, supporting smooth semantic transitions and tolerating moderate deviations from the training distribution. Please refer to Appendix D for more details.

## 5 CONCLUSION

In this work, we revisit the latent diffusion paradigm and identify the absence of a semantically discriminable latent structure as a key factor limiting training and inference efficiency. To address

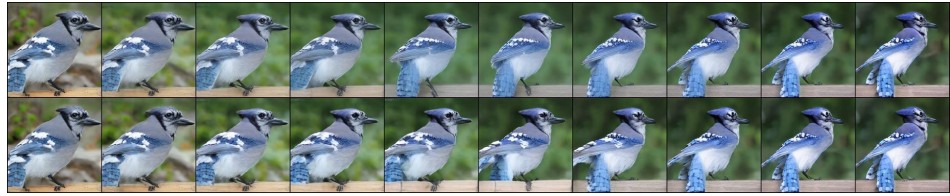

Figure 7: **Visualization of interpolation using SVG.** The first row shows direct linear interpolation, while the second row presents spherical linear interpolation.

this, we propose SVG, a latent diffusion model without variational autoencoders, which enriches frozen DINO features with residual features capturing fine-grained perceptual details. This unified feature space supports diverse core vision tasks, enabling faster diffusion training, efficient few-step sampling, and improved generative quality. These results position SVG as a promising approach toward a single representation that unifies generation with other diverse visual tasks.

**Limitations and future work.** In this work, we explore the potential of using VFM features to construct a latent space for diffusion training. Experiments confirm its feasibility, though further improvements remain, such as reducing the dimensionality of SVG features or refining the residual encoder to enhance efficiency and generative quality. We also find that classifier-free guidance is less effective in our framework, indicating the need for better alternatives. Beyond current experiments, the potential of SVG on larger datasets, higher resolutions, and more challenging T2I/T2V tasks remains underexplored. We are investigating its application to text-to-image generation, and given the strong grounding ability of SVG features, we believe it also holds great promise for visual editing.

ACKNOWLEDGMENTS

This work was supported in part by the National Natural Science Foundation of China under Grant 62321005, Grant 62336004, Grant 62125603, and in part by the Beijing Natural Science Foundation under Grant No. L247009.

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

# A   MORE IMPLEMENTATION DETAILS

Table 5: Hyperparameter setup.

|  | SVG-B | SVG-L | SVG-XL | SiT-XL |
|---|---|---|---|---|
| **Architecture** | | | | |
| Input dim. | $16 \times 16 \times 384$ | $16 \times 16 \times 384$ | $16 \times 16 \times 384$ | $32 \times 32 \times 4$ |
| Num. layers | 12 | 24 | 28 | 28 |
| Hidden dim. | 768 | 1024 | 1152 | 1152 |
| Num. heads | 12 | 16 | 16 | 16 |
| Base-encoder | DINOv3-s16p | DINOv3-s16p | DINOv3-s16p | SD-VAE |
| **Optimization** | | | | |
| Batch size | 256 | 256 | 256 | 256 |
| Optimizer | AdamW | AdamW | AdamW | AdamW |
| lr | 0.0001 | 0.0001 | 0.0001 | 0.0001 |
| $(\beta_1, \beta_2)$ | (0.9, 0.999) | (0.9, 0.999) | (0.9, 0.999) | (0.9, 0.999) |
| **Interpolants** | | | | |
| $\alpha_t$ | $1 - t$ | $1 - t$ | $1 - t$ | $1 - t$ |
| $\sigma_t$ | $t$ | $t$ | $t$ | $t$ |
| Training objective | v-prediction | v-prediction | v-prediction | v-prediction |
| Sampler | Euler | Euler | Euler | Euler |
| Sampling steps | 25 | 25 | 25 | 250 |
| Guidance | - | - | 1.55 | 1.5 |

**Hypermarameters.** We report the hyperparameters for architecture, optimization and interpolants in Table 5.

**Computing resources.** We train our reconstruction and generation experiments on $8\times$H100 GPUs.

**Sampler.** For a fair comparison, we adopt Euler's method to solve the ODE for image generation. As a first-order sampler, the number of steps in Euler's method directly corresponds to the number of function evaluations (NFE).

**Classifier-free guidance.** In the main text, we report results both with and without classifier-free guidance. To reduce the uncertainty in the initial velocity prediction, we adopt zero-init (Fan et al., 2025), which skips the first step. We report the FID50K with cfg 1.5 in our paper.

# B   FINETUNING DETAILS ON DOWNSTREAM TASKS

We evaluate the SVG encoder against the DINOv3 encoder on three representative downstream tasks: ImageNet-1K (Deng et al., 2009; Russakovsky et al., 2015) classification, ADE20K (Zhou et al., 2019) semantic segmentation, and NYUv2 (Nathan Silberman & Fergus, 2012) depth estimation. For all tasks, the encoder remains frozen, and only lightweight decoders are trained.

**Image classification.** This task requires assigning each image to a single class. We report Top-1 and Top-5 accuracies. A linear classifier is placed on top of the encoder output to map features to class scores. Input images are randomly resized and cropped to $256 \times 256$. The classifier is trained for 30 epochs with the AdamW optimizer (Loshchilov & Hutter, 2017), using a global batch size of 3072, an initial learning rate of $5e - 4$, and weight decay of $1e - 2$.

**Semantic segmentation.** The goal of semantic segmentation is to produce dense per-pixel predictions. We use mean Intersection-over-Union (mIoU) and mean Accuracy (mAcc) as evaluation metrics. The decoder adopts an FPNHead implementation in *mmsegmentation*, applied to the single-scale encoder feature. The training strategy generally follows (Zhao et al., 2023). Images are randomly resized and cropped to $512 \times 512$ before being fed to the network. Optimization is performed with AdamW (Loshchilov & Hutter, 2017) at a learning rate of $8e - 5$, weight decay of $1e - 3$, and 1,500 warm-up steps. A polynomial scheduler with power 0.9 and a minimum learning rate of $1e - 6$ is used. Training runs for 8,000 iterations. During inference, we adopt sliding-window evaluation with $512 \times 512$ crops and a stride of $341 \times 341$.

**Depth estimation.** Depth estimation aims to regress pixel-wise depth values for input images. We report Absolute Relative Error (A.Rel) and RMSE as evaluation metrics. During training, images are randomly cropped to $480 \times 480$. The model is optimized for 25 epochs using the AdamW (Loshchilov & Hutter, 2017) optimizer with a batch size of 24 and a learning rate of $5e - 4$. The decoder head and other hyperparameters follow the setup in (Xie et al., 2022a). At test time, we use both horizontal flipping and sliding-window inference.

## C   EDITING DETAILS

For editing experiments, we adopt an SDEdit-style (Meng et al., 2021) procedure with trajectory inversion to preserve spatial and semantic consistency. Specifically, given an input image, we first invert it to the diffusion latent space and record its noisy trajectory up to a target timestep $t_{\text{edit}}$. The inversion trajectory provides a reference for the preserved regions during subsequent editing, ensuring that unchanged areas remain faithful to the original content. At $t_{\text{edit}}$, we apply a binary spatial mask on the latent feature maps: the masked regions are replaced with Gaussian noise while the unmasked regions retain their inverted latents. Two editing strategies are considered: (i) preserving the content outside the red box and editing the inside, and (ii) preserving the inside while editing the outside. To achieve smooth transitions, the mask is softened with a 2D Gaussian blur and dynamically faded during denoising. From this initialization, we resume forward sampling under the new class condition using Euler's method with 100 steps, classifier-free guidance scale $4.0$, and timestep shift $0.4$. Finally, the SVG decoder reconstructs the full-resolution image. This inversion-guided process ensures that edits are spatially coherent, semantically aligned with the target class, and smoothly integrated with preserved regions.

## D   LATENT SPACE INTERPOLATION TEST

To assess the continuity of the proposed SVG feature space, we perform a latent space interpolation test. We randomly sample two noise vectors $\boldsymbol{x}_T^0$ and $\boldsymbol{x}_T^1$ from the standard Gaussian distribution and generate interpolants conditioned on the class embedding. Visual results are presented in Figures 8 and 9.

For linear interpolation, we compute

$$\boldsymbol{x}_T^\lambda = (1 - \lambda)\boldsymbol{x}_T^0 + \lambda \boldsymbol{x}_T^1, \qquad \lambda \in [0, 1] \tag{7}$$

And for spherical linear interpolation (slerp), we use

$$\boldsymbol{x}_T^\lambda = \frac{\sin((1 - \lambda)\theta)}{\sin \theta} \, \boldsymbol{x}_T^0 + \frac{\sin(\lambda\theta)}{\sin \theta} \, \boldsymbol{x}_T^1, \qquad \lambda \in [0, 1] \tag{8}$$

where $\theta = \arccos\left( \frac{(\boldsymbol{x}_T^0)^\top \boldsymbol{x}_T^1}{\|\boldsymbol{x}_T^0\|\|\boldsymbol{x}_T^1\|} \right)$.

Spherical interpolation (slerp) is theoretically preferable because it better preserves vector norms and therefore is less likely to produce interpolants that deviate strongly from the distributions seen during training. Empirically, the SVG-Autoencoder outputs vary smoothly with $\lambda$ under slerp. Remarkably, even with direct linear interpolation—whose samples need not follow the Gaussian prior and thus are out-of-distribution relative to training—the generated images remain natural and high-quality in our method. By contrast, VAE-based counterparts degrade under such linear interpolations. These results demonstrate that the proposed SVG feature space exhibits strong continuity and robustness: its geometry supports smooth semantic transitions and the trained diffusion model tolerates moderate deviations in the input noise distribution.

## E   ADDITIONAL COMPARISONS OF DIFFERENT LATENT SPACE

As shown in Figure 10, removing distribution alignment noticeably weakens the semantic discriminability of the SVG feature space, which subsequently leads to degraded generative performance (Table 4). This demonstrates that distribution alignment is essential for maintaining both a well-structured latent space and strong generation quality.

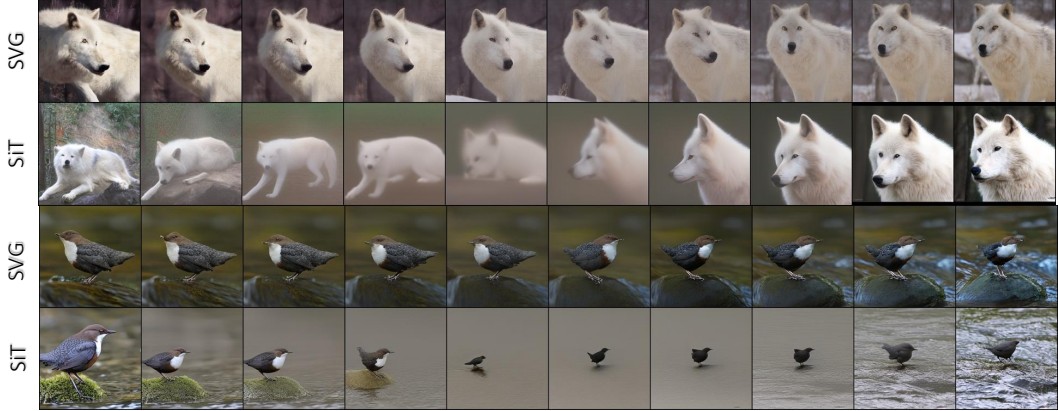

Figure 8: **Visualization of linear interpolation.** Two noise vectors are randomly sampled and linearly interpolated under the same class embedding.

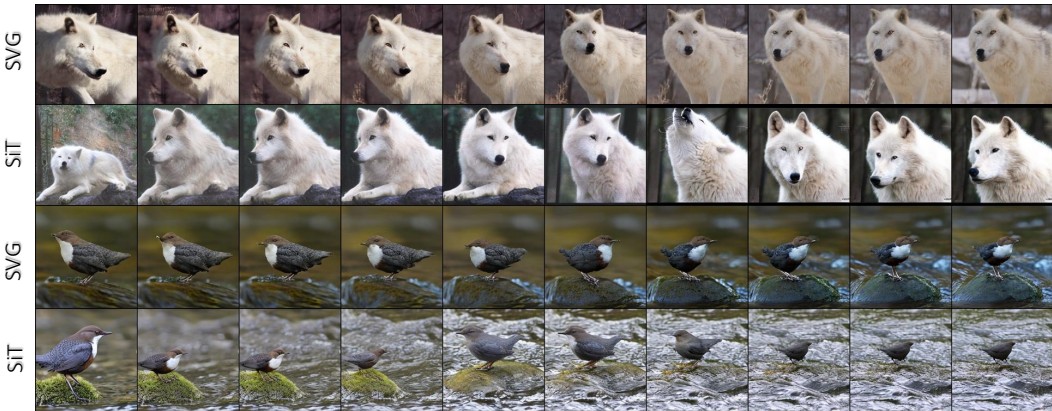

Figure 9: **Visualization of spherical linear interpolation.** Two noise vectors are randomly sampled and spherical linearly interpolated under the same class embedding.

To better understand how the choice of encoder feature space affects diffusion-based generation, we conducted a series of experiments using features extracted from different intermediate layers of the DINOv3 encoder. For each selected layer, we trained diffusion models under identical settings and evaluated both reconstruction quality (rFID) and generative quality (gFID). In addition, supplementary t-SNE visualizations in Figure 10 are provided to further illustrate the distributional characteristics of these feature spaces. The results in Table 6 show a consistent trend: shallower layers lead to inferior generative performance even when reasonable reconstruction can be achieved. This behavior is expected, as shallow layers primarily capture low-level textures and fine local details, but lack semantic abstraction. In contrast, deeper layers encode higher-level semantics that are more suitable for generative modeling. These observations offer practical guidance for selecting the most suitable feature space when training diffusion models.

## F    TOKENIZER SIZE AND INFERENCE EFFICIENCY

We report the parameter counts and computational cost of the tokenizer to provide a clearer comparison across different autoencoder designs. As shown in Table 7, SVG Autoencoder exhibits a parameter scale and end-to-end latency comparable to prior baselines, with the latency differences being minor relative to the computational cost of the diffusion backbone. A notable advantage of SVG lies in its encoder-side efficiency. The encoder FLOPs are substantially lower than those of existing VAE-based designs, while its throughput is significantly higher. This indicates meaningful scalability benefits when deployed in large-batch or large-scale computation scenarios.

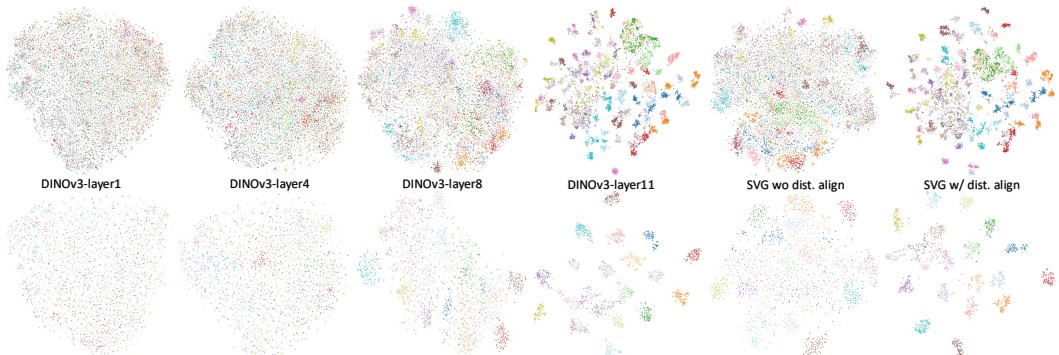

Figure 10: **The t-SNE visualization of different visual latent spaces.** We use two datasets for feature extraction: The first row has 100 ImageNet classes with 100 samples per class, and the second row has 20 ImageNet classes with 100 samples per class. Each class is represented by a distinct color.

Table 6: **Performance of diffusion models trained on encoder intermediate layers.** Training Epochs indicate the number of epochs used to train the SVG Autoencoder. The gFID metrics are obtained after 200K training steps of the SVG-B diffusion model, without classifier-free guidance.

| Layer | Training Epochs | rFID ↓ | gFID ↓ |
|:-----:|:---------------:|:------:|:------:|
| 4     | 5               | 0.792  | 73.45  |
| 4     | 20              | 0.447  | –      |
| 8     | 5               | 1.309  | 36.63  |
| 8     | 20              | 0.859  | –      |
| 11    | 5               | 1.753  | 27.00  |
| 11    | 20              | 0.992  | –      |

Table 7: **Tokenizer parameter counts and inference efficiency.** Throughput is measured using batch size 64, while latency corresponds to single-image inference (batch size = 1).

| Module | #Params | GFLOPs | Latency (ms/img) | Throughput (imgs/s) |
|:-------|:-------:|:------:|:----------------:|:-------------------:|
| SDVAE-Encoder  | 34.16M | 135.59 | 7.54  | 224.22  |
| VAVAE-Encoder  | 28.41M | 69.21  | 6.45  | 332.87  |
| DINOv3-s16plus | 28.70M | 7.48   | 11.49 | 2118.88 |
| **SVG-Encoder**    | **39.89M** | **10.33**  | **14.28** | **1555.07** |
| SDVAE-Decoder  | 49.49M | 310.62 | 13.76 | 114.26  |
| VAVAE-Decoder  | 41.42M | 126.56 | 9.43  | 198.60  |
| **SVG-Decoder**    | **43.08M** | **126.94** | **9.33**  | **199.68**  |
| SiT-XL         | –      | –      | 426.48 | 10.56  |

## G    FURTHER ANALYSIS OF SVG GENERATION

We present PCA visualizations of the feature maps in Figure 11 and Figure 12, following the approach of DINOv3 (Siméoni et al., 2025). Compared to the vanilla VAE-based DiT model, which tends to produce noisy feature maps, especially at large timesteps, SVG yields cleaner and more structured representations. The hidden states of SVG exhibit more discriminative characteristics, which are beneficial for both generation quality and downstream tasks.

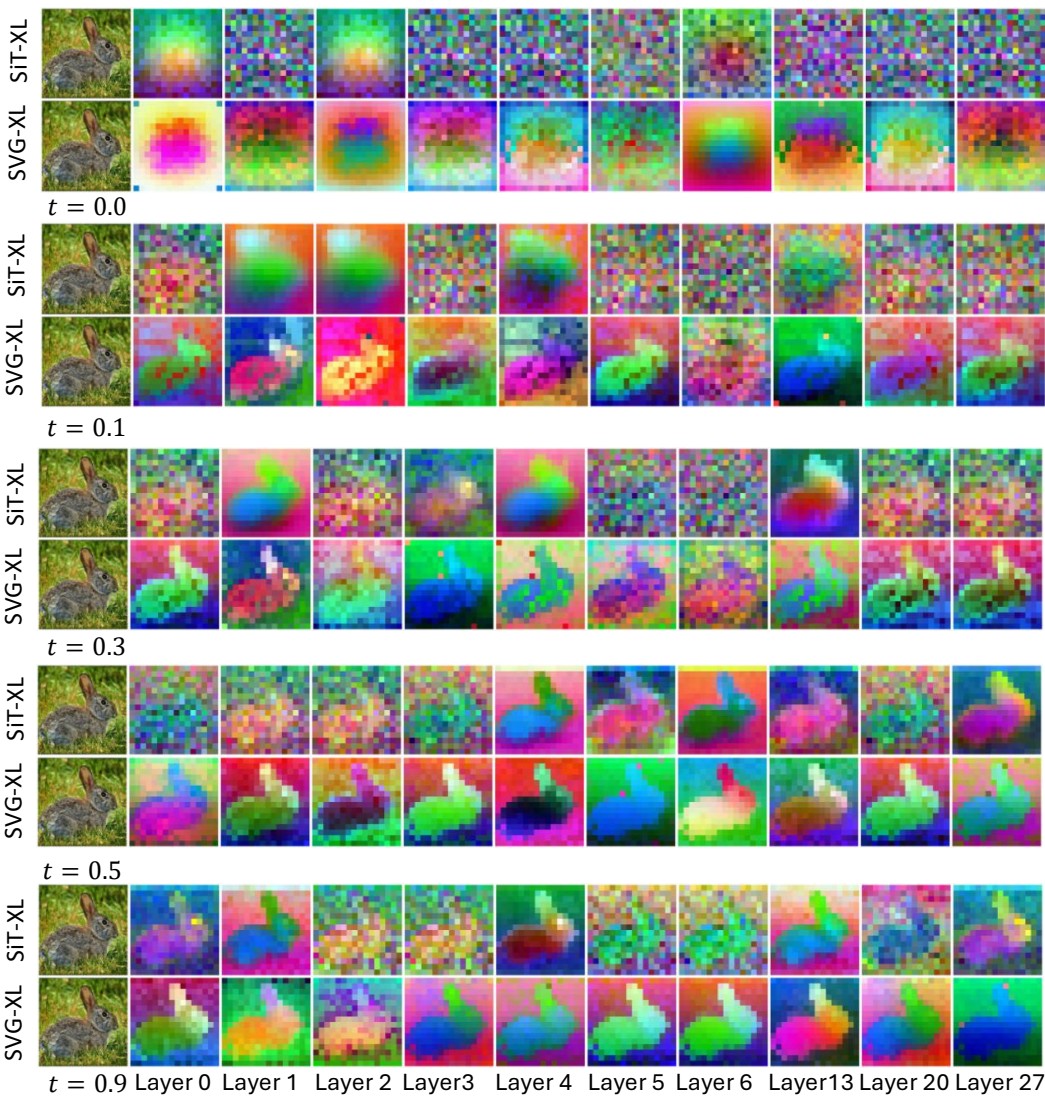

Figure 11: **PCA visualizations of feature maps.** SVG shows cleaner feature maps, while the VAE-Diffusion model tends to show noisy feature maps, particularly for large $t$.

## H  SCALING TO HIGHER RESOLUTION

To further assess our model's high-resolution generative capability, we conducted additional reconstruction and generation experiments at 512×512 and 1024×1024. For the SVG Autoencoder, we first resumed from the checkpoint trained at 256×256 for 40 epochs, then continued training at 512×512 for 5 epochs, and finally performed one epoch of finetuning at 1024×1024. For the diffusion model, we resumed from the SVG-XL checkpoint trained at 256×256 for 300 epochs, subsequently finetuned it at 512×512 for 25 epochs, and completed an additional epoch at 1024×1024. The corresponding visual results are shown in Figures 15 to 18, confirming that our method scales to higher resolutions without requiring any architectural modifications.

## I  MORE QUALITATIVE RESULTS

We provide additional qualitative results of SVG-XL on ImageNet $256 \times 256$. Randomly selected samples are shown in Figure 19, while uncurated generations for specific classes are presented

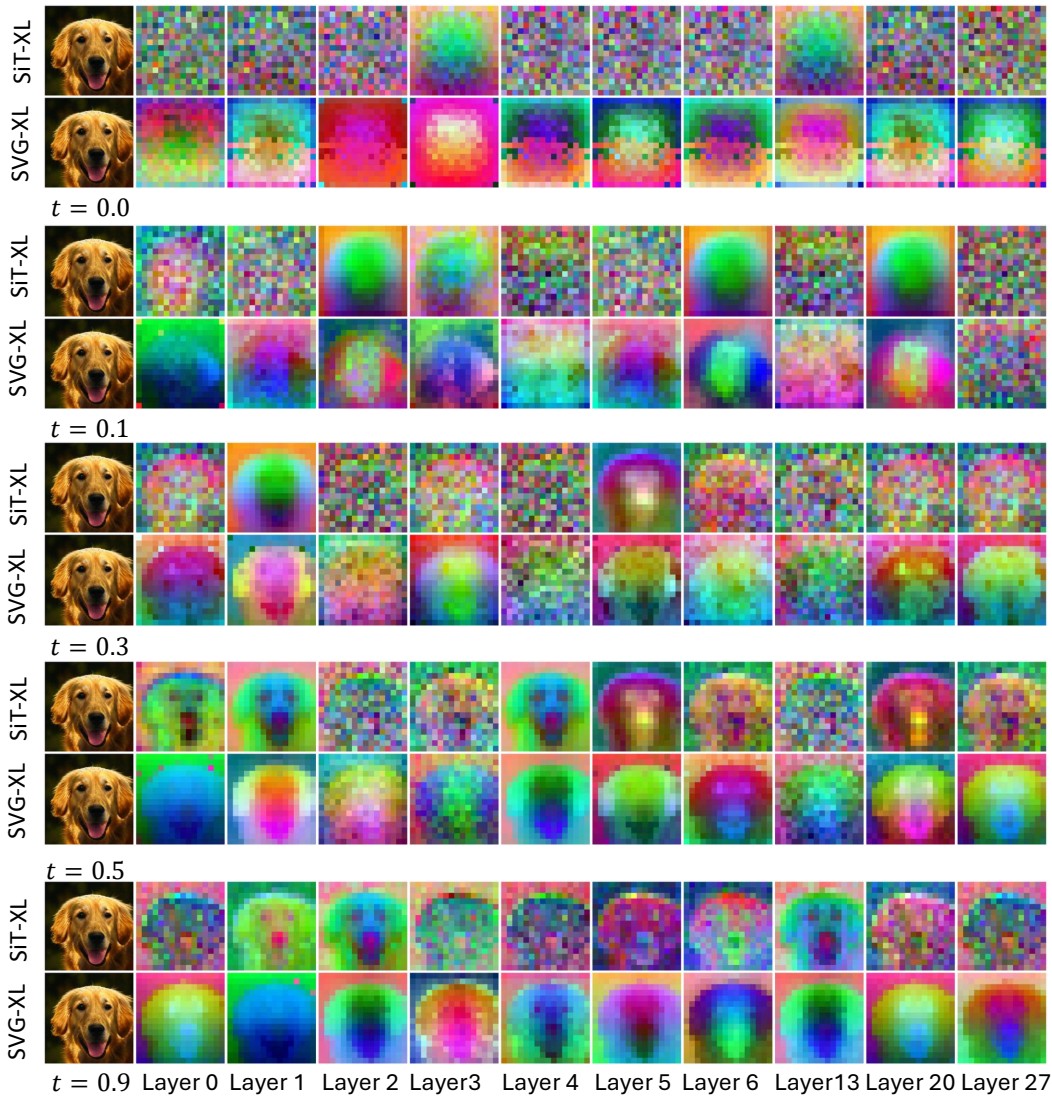

Figure 12: **PCA visualizations of feature maps.** SVG shows cleaner feature maps, while the VAE-Diffusion model tends to show noisy feature maps, particularly for large $t$.

in Figures 20 and 22 to 33. These results further demonstrate the diversity and visual quality of the proposed approach.

## J    DESCRIPTION OF PRETRAINED VISUAL ENCODERS

**MAE (He et al., 2021).** Masked Autoencoders (MAE) is a self-supervised pre-training framework. Its core principle lies in reconstructing randomly masked image patches from the remaining visible ones. MAE achieves efficient training while forcing the model to capture high-level semantic information.

**DINO (Zhang et al., 2022).** DINO is a self-supervised method leveraging self-distillation without using any human-provided labels. It trains two neural networks (a student and a teacher) on different augmented views of the same image. Specifically, the teacher's parameters are an exponential moving average of the student's, and the student is optimized to align its output with the teacher's. By eliminating the need for labels or negative sample mining, DINO learns highly discriminative features that exhibit strong transferability on various downstream perception tasks.

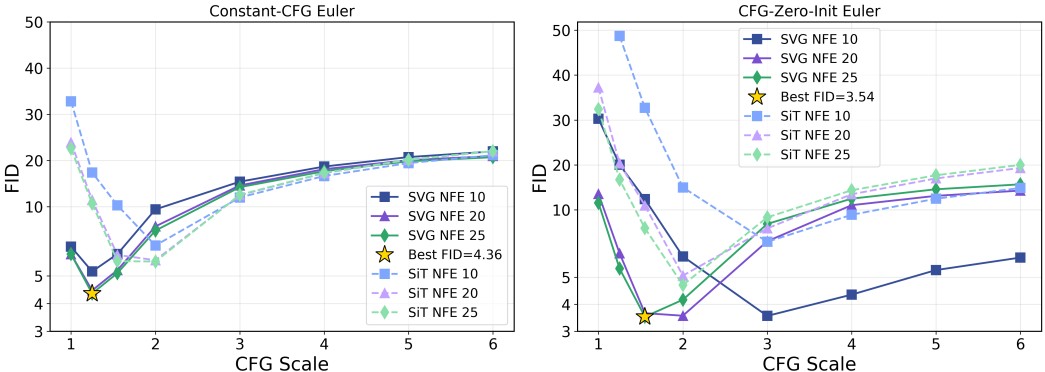

Figure 13: **Impact of classifier-free guidance on SVG generation performance.** Evaluated using SVG-XL trained at 256x256 for 80 epochs.

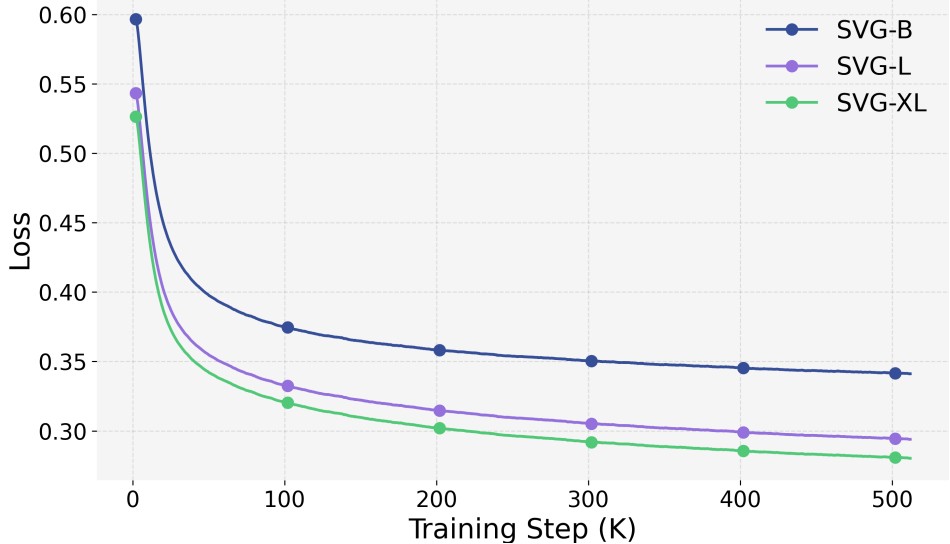

Figure 14: **Training curves of SVG.** All model scales (B, L, XL) demonstrate similarly stable convergence behavior.

**DINOv2 (Oquab et al., 2023).** DINOv2 systematically improves training, data, efficiency, and model distillation. It combines DINO's image-level contrastive loss with iBOT's patch-level masked image modeling and introduces KoLeo regularization, enabling learning of both global and local representations.

**DINOv3 (Siméoni et al., 2025).** DINOv3 builds upon predecessors by introducing several key improvements to enhance self-supervised visual representation learning, particularly for dense features and large-scale training. It scales both the dataset and model size. A novel Gram Anchoring strategy stabilizes patch-level representations during long training, producing higher-quality dense feature maps. Additionally, high-resolution post-training and efficient knowledge distillation allow compressing the 7B model into smaller variants while retaining strong performance.

**CLIP (Radford et al., 2021).** CLIP is a multi-modal pre-training framework that aligns visual and linguistic representations. It jointly trains a visual encoder and a text encoder via contrastive learning. Given image-text pairs, the model maximizes the similarity between matching pairs while minimizing similarity between non-matching ones. This aligns the image and text embedding spaces, enabling zero-shot transfer to downstream tasks. CLIP's versatility lies in its ability to generalize to unseen concepts without task-specific fine-tuning.

**SigLIP (Zhai et al., 2023).** SigLIP improves upon CLIP by replacing the softmax-based contrastive loss with a pairwise sigmoid cross-entropy loss. This modification makes it more scalable to massive datasets. SigLIP maintains strong alignment between vision and language while being more efficient, achieving superior performance compared to CLIP on zero-shot and fine-tuned benchmarks.

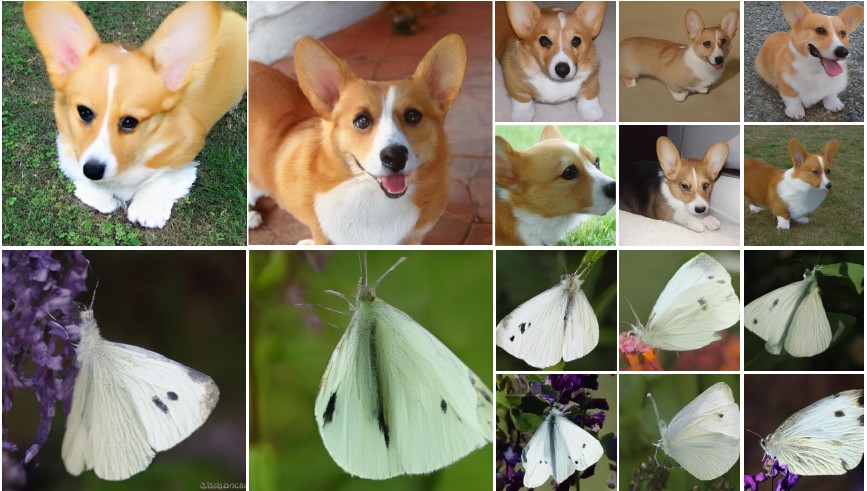

Figure 15: **512x512 Generation Results** of SVG-XL. We use classifier-free guidance with w = 4.0

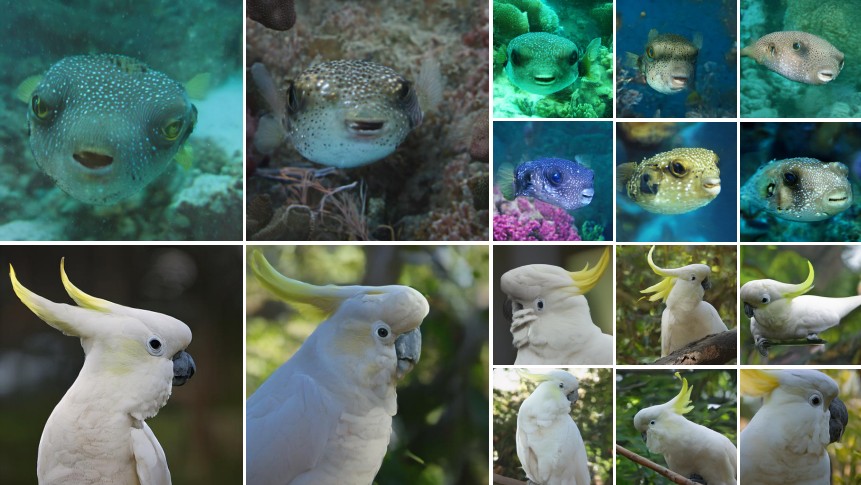

Figure 16: **512x512 Generation Results** of SVG-XL. We use classifier-free guidance with w = 4.0

**SigLIP2 (Tschannen et al., 2025).** SigLIP2 represents a systematic upgrade over SigLIP, evolving from a single-loss contrastive framework into a unified training recipe that integrates decoder-based pretraining, self-supervised objectives, and new engineering techniques. SigLIP2 introduces a transformer decoder to enhance local detail understanding via captioning and referring expression tasks, while additional self-distillation and masked prediction losses significantly improve dense prediction performance. It further extends multilingual coverage by training on larger datasets.

## K    STATEMENT ON LLM ASSISTANCE

Parts of the manuscript were polished for clarity and readability using ChatGPT and DeepSeek. The authors are solely responsible for the technical content and conclusions of this work.

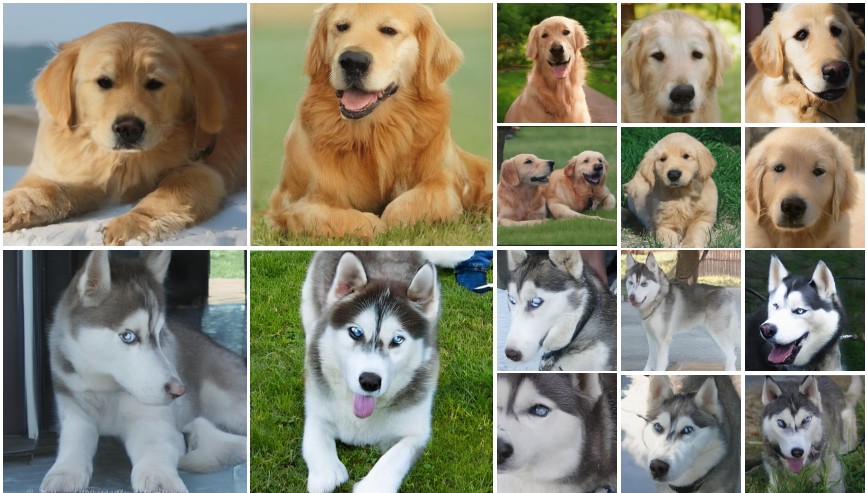

Figure 17: **512x512 Generation Results** of SVG-XL. We use classifier-free guidance with w = 4.0

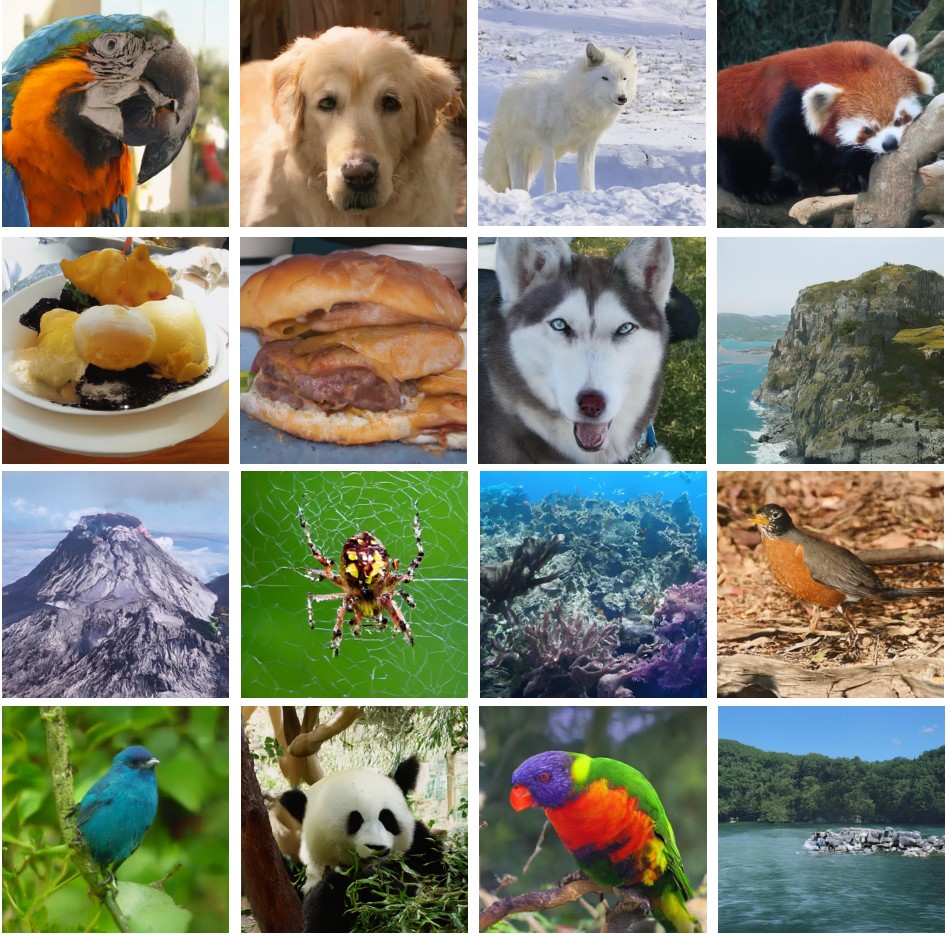

Figure 18: **1024x1024 Generation Results** of SVG-XL. We use classifier-free guidance with w = 4.0

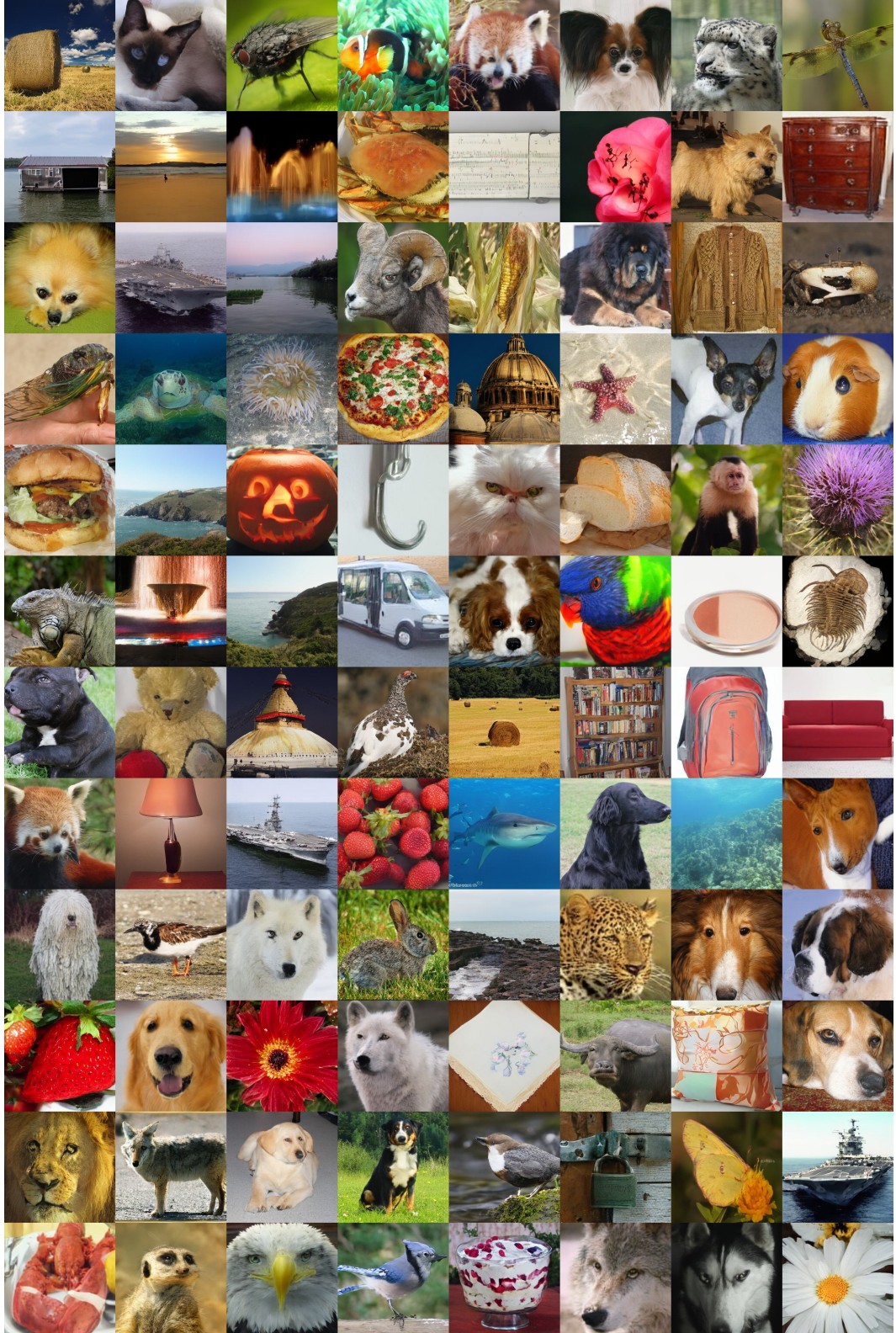

Figure 19: **Random samples from SVG-XL on ImageNet 256×256.** We use a classifier-free guidance scale of 4.0

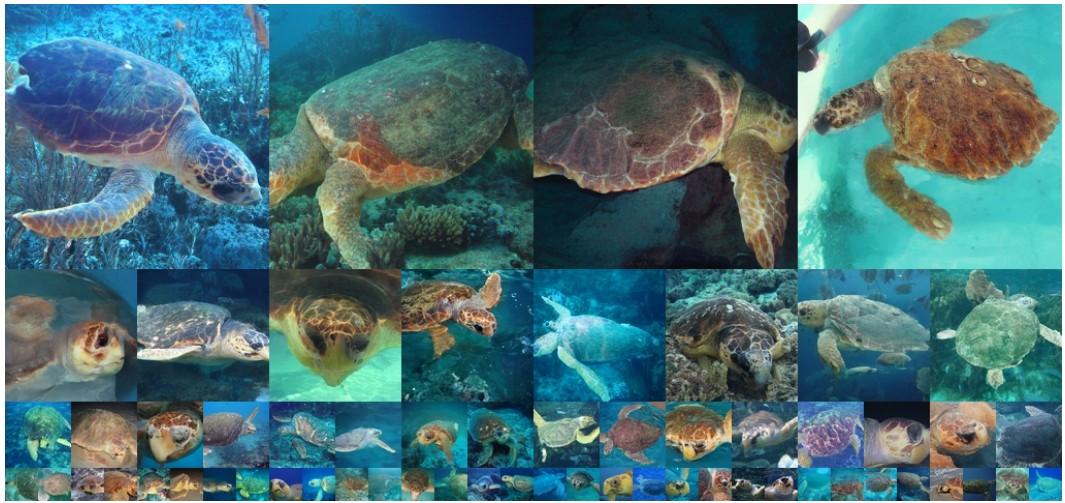

Figure 20: **Uncurated generation results of SVG-XL**. We use classifier-free guidance with w = 4.0. Class label = 33.

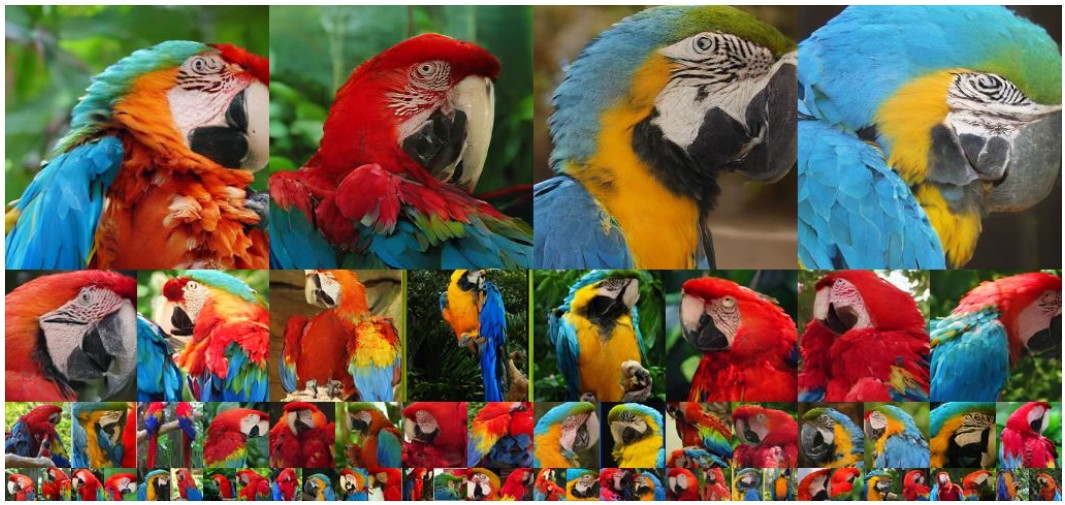

Figure 21: **Uncurated generation results of  SVG-XL**. We use classifier-free guidance with w = 4.0. Class label = 88.

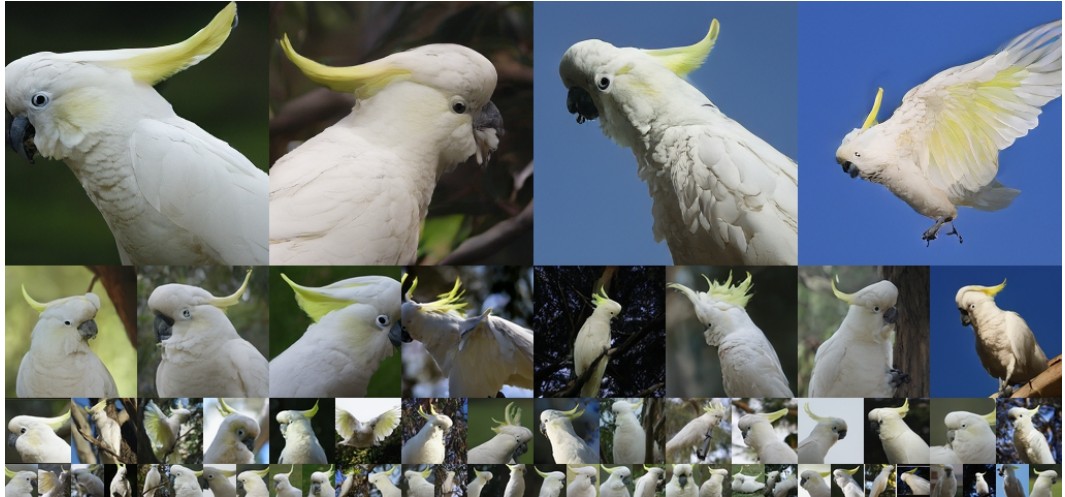

Figure 22: **Uncurated generation results of SVG-XL**. We use classifier-free guidance with w = 4.0. Class label = 89.

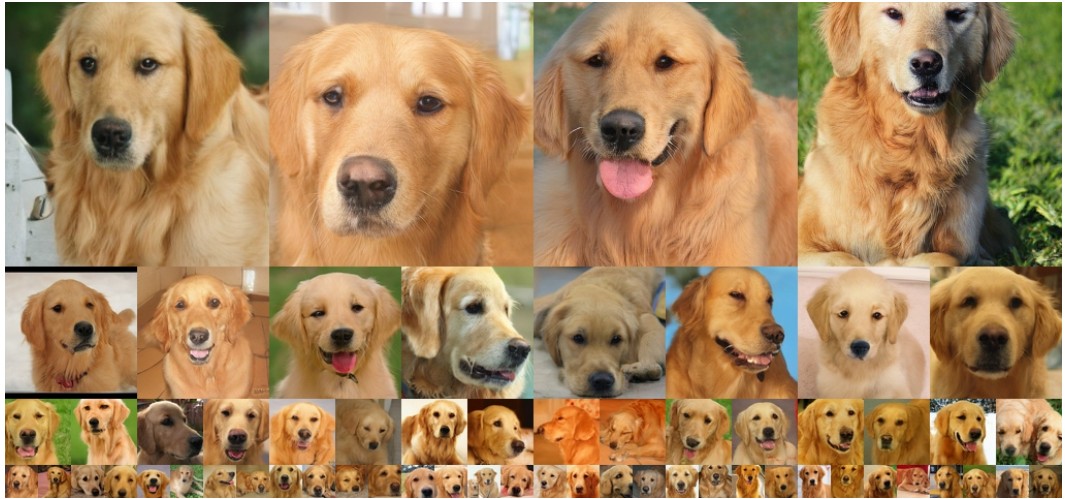

Figure 23: **Uncurated generation results of SVG-XL**. We use classifier-free guidance with w = 4.0. Class label = 207.

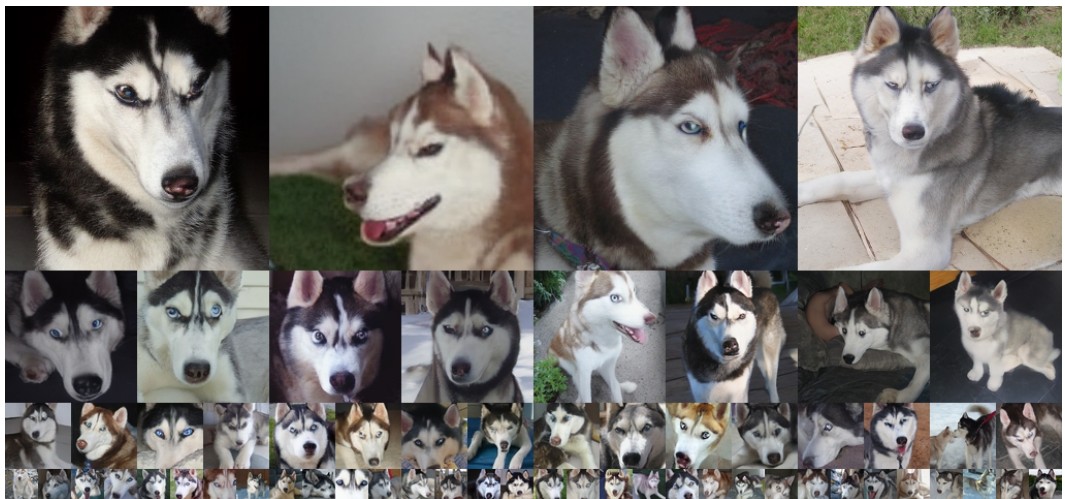

Figure 24: **Uncurated generation results of SVG-XL**. We use classifier-free guidance with w = 4.0. Class label = 250.

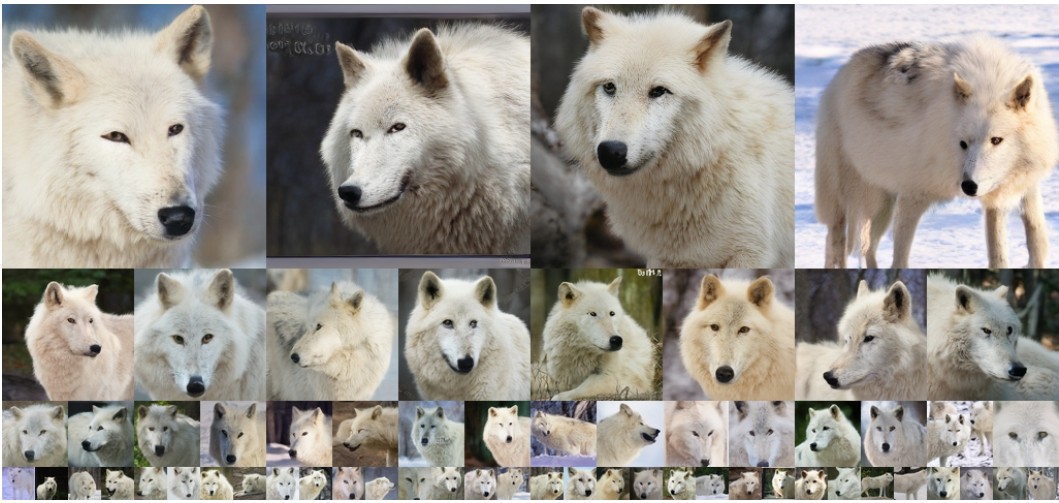

Figure 25: **Uncurated generation results of SVG-XL**. We use classifier-free guidance with w = 4.0. Class label = 270.

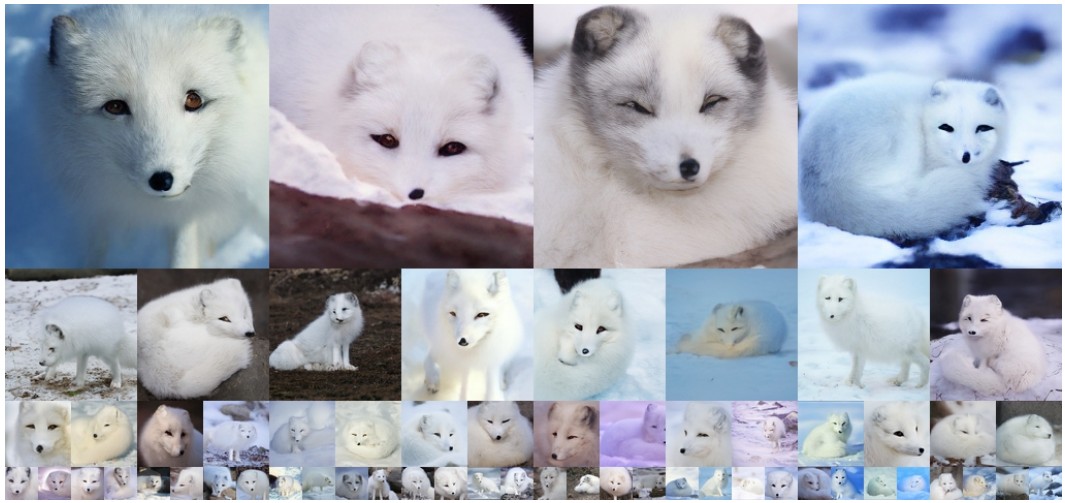

Figure 26: **Uncurated generation results of SVG-XL**. We use classifier-free guidance with w = 4.0. Class label = 279.

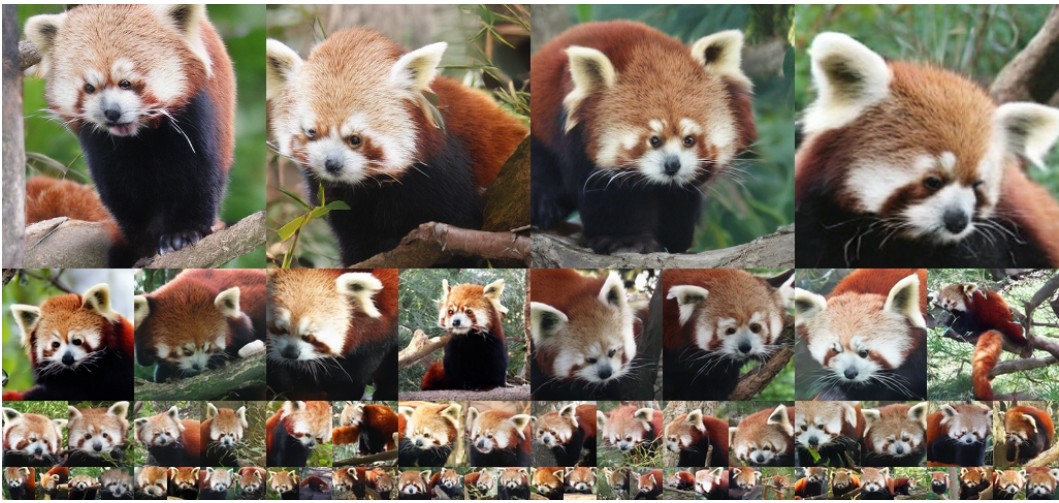

Figure 27: **Uncurated generation results of SVG-XL**. We use classifier-free guidance with w = 4.0. Class label = 387.

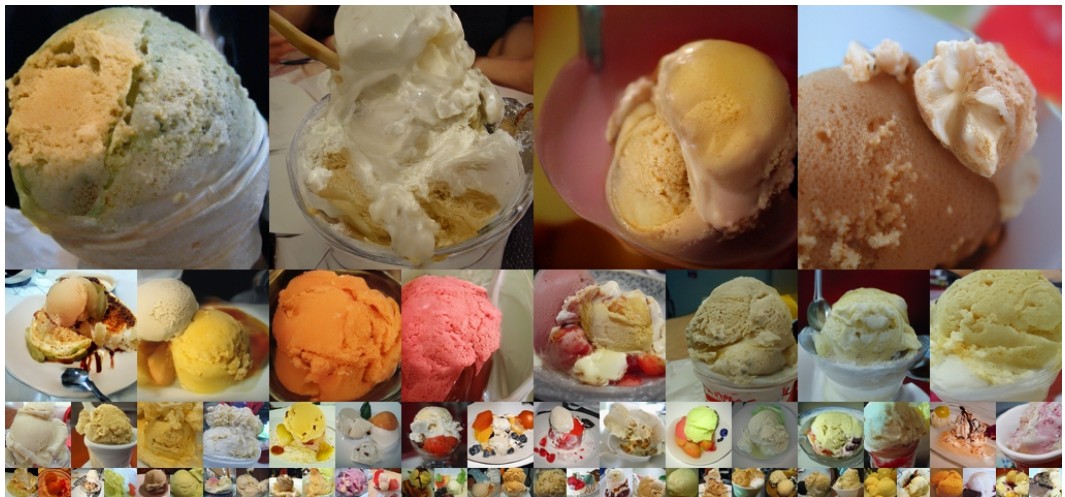

Figure 28: **Uncurated generation results of SVG-XL**. We use classifier-free guidance with w = 4.0. Class label = 928.

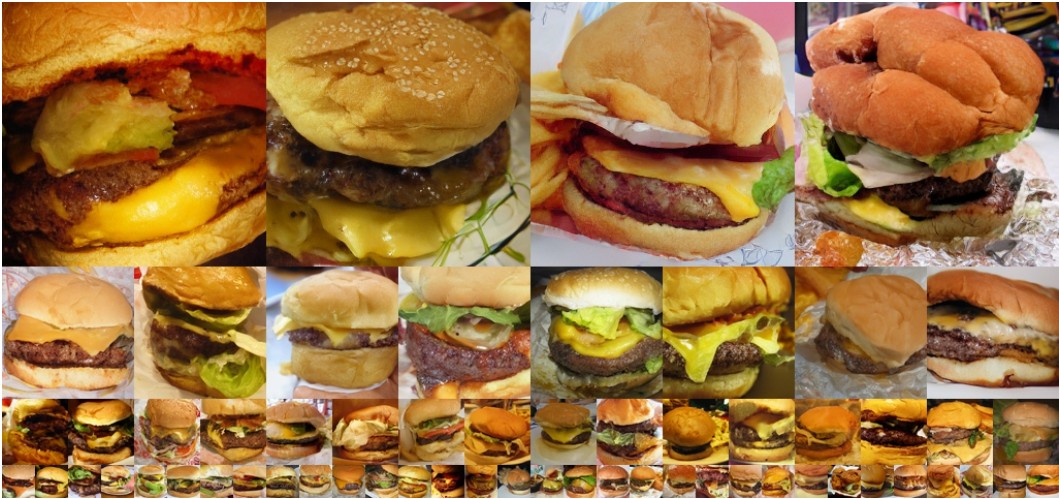

Figure 29: **Uncurated generation results of SVG-XL**. We use classifier-free guidance with w = 4.0. Class label = 933.

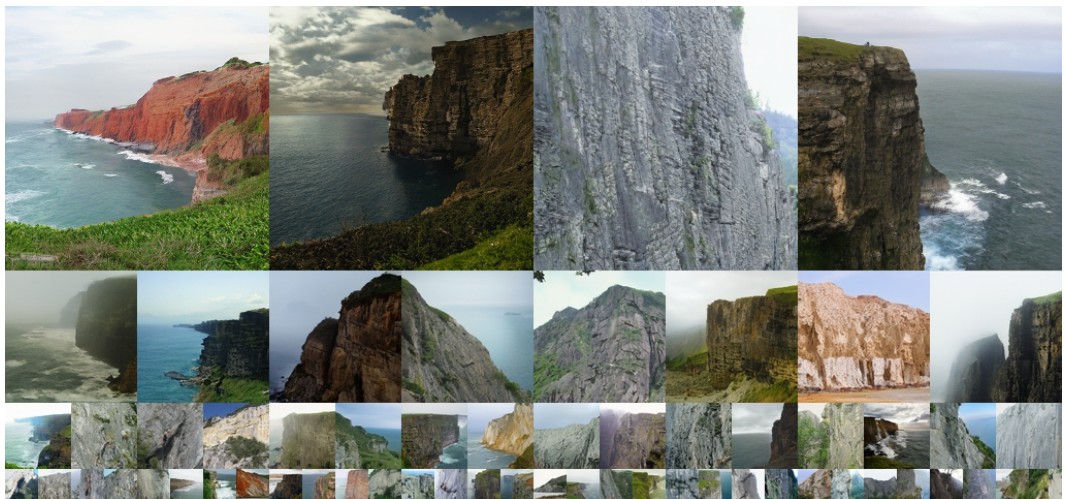

Figure 30: **Uncurated generation results of SVG-XL**. We use classifier-free guidance with w = 4.0. Class label = 972.

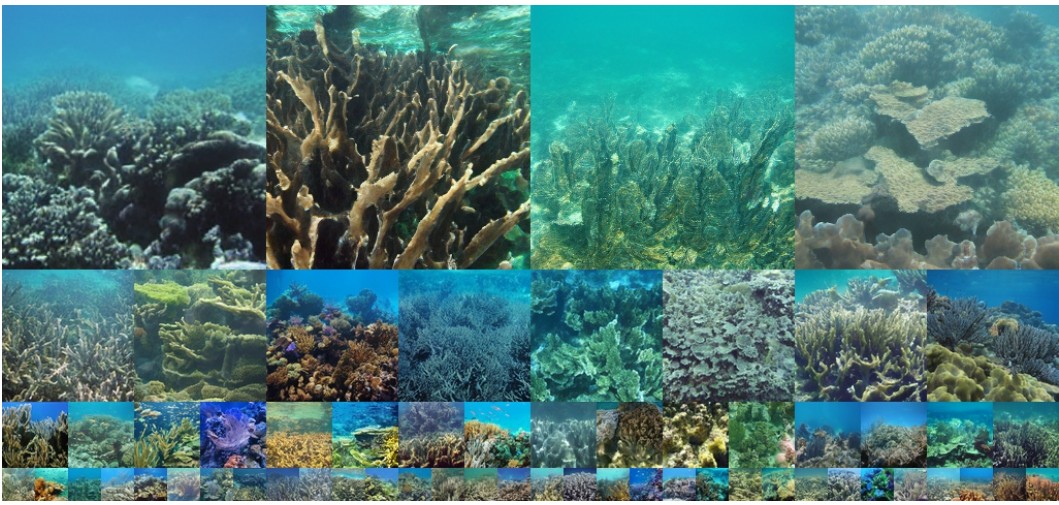

Figure 31: **Uncurated generation results of SVG-XL**. We use classifier-free guidance with w = 4.0. Class label = 973.

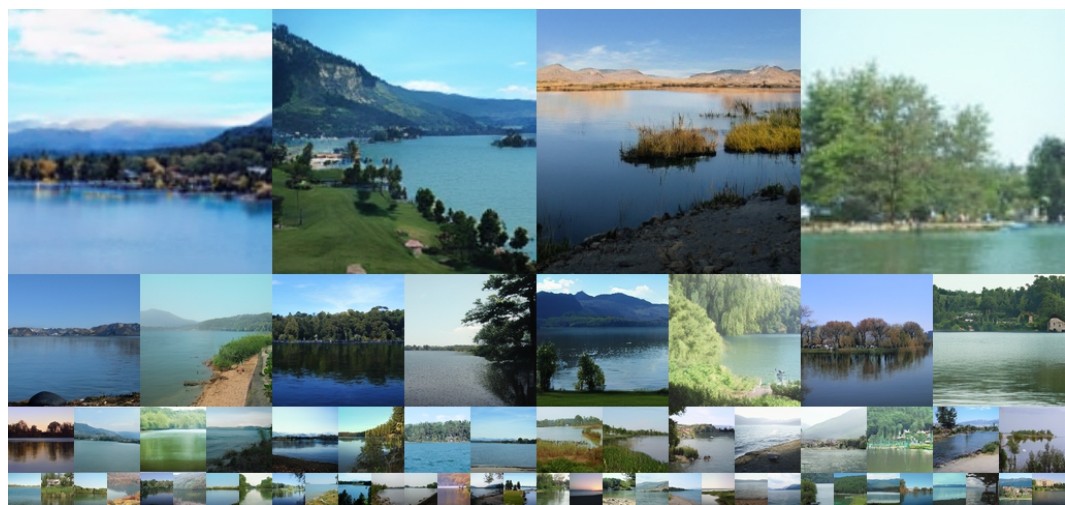

Figure 32: **Uncurated generation results of SVG-XL**. We use classifier-free guidance with w = 4.0. Class label = 975.

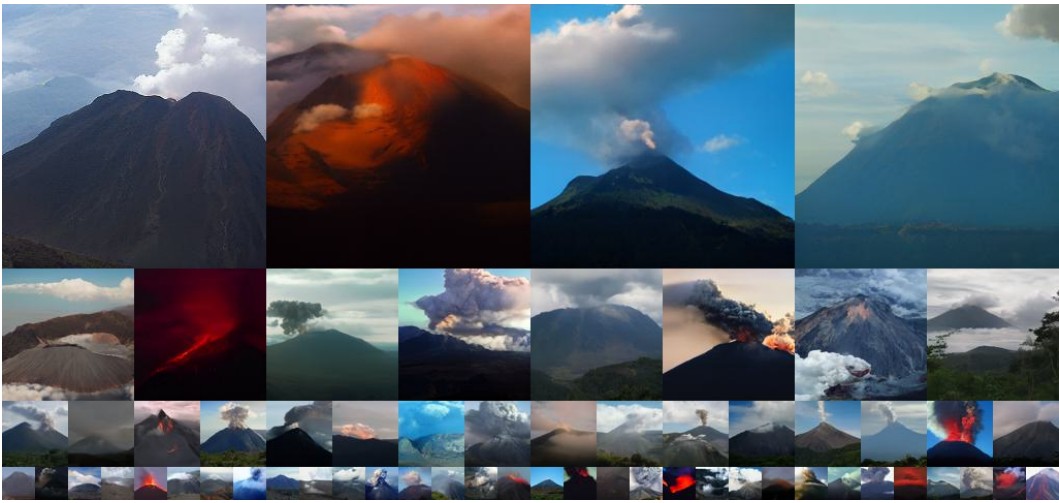

Figure 33: **Uncurated generation results of SVG-XL**. We use classifier-free guidance with w = 4.0. Class label = 980.

