# OpenReview forum: "Latent Diffusion Model without Variational Autoencoder"
_ICLR.cc/2026/Conference — ICLR 2026 Poster_

### Official Review · Reviewer_go1M · 2025-10-28

**Soundness:** 3
**Presentation:** 3
**Contribution:** 3
**Rating:** 6
**Confidence:** 4

**Summary:**

The authors introduce SVG - a novel latent diffusion model without variational autoencoders. The method leverages features from a DINO model while a residual branch captures features useful for high fidelity generation. The authors claim that the method supports accelerated training, few step sampling and improved generation quality which is supported by extensive empirical comparisons with prior baselines.

**Strengths:**

1. **Simplicity of the approach**: The proposed pipeline is simple: Training diffusion models in a more discriminative feature space while augmenting it with residual generative feature spaces.

2. **Empirical results**: I like the ablation experiments in Section 4.3 which clearly highlight the contributions of the core components of the method. Similarly the main results presented in Section 4.2 demonstrate the proposed method has better training efficiency than competing baselines. I also like Fig. 4b which presents a nice intuitive explanation of the impact of mode separation on diffusion models training. If the modes are mixed, any diffusion or flow model will have trouble learning the score directions while a clear mode separation enhances learning.

**Weaknesses:**

1. In the abstract (line 014): The authors mention slow inference as one of the drawbacks of Latent diffusion models (LDMs). This claim is a bit shaky as the latent space dimension is often much less than the original image (compare 3x512x512 original image size to 64x32x32 in StableDiffusion 1.5). Therefore inherently LDMs support fast sample generation than a diffusion model trained solely in the pixel space. Moreover, in my personal experience, with fast samplers like DPM solver, good quality sample generation takes around 20-50 diffusion steps and is quite fast. Therefore I’m not sure how valid this claim is in practice.

2. **Presentation**: In current Fig.1, the top row in the figure showing the block diagram for the architecture is a bit uninformative in terms of different color coding schemes. Also the figure caption provides no meaningful information and should be revised accordingly to make it more descriptive.

3. **Related Work**: In line 67 the authors claim that VAE latent spaces inherently lack semantic separability. Can the authors provide a few citations to support this claim here? There has been a lot of work in disentangling the VAE representations starting from Beta-VAEs [Higgins et al]. Similarly a citation for DINOv3 is missing in Line 73 where it is first introduced. There is some related work on combining VAEs with Diffusion models which is missing from the main text to which these advances can be potentially applied too and thus worth discussing in the related work section. For instance:

a. Score-based Generative Modeling in Latent Space, Vahdat et al.

b. DiffuseVAE: Efficient, Controllable and High-Fidelity Generation from Low-Dimensional Latents, Pandey et al.

c. Diffusion Autoencoders: Toward a Meaningful and Decodable Representation, Preechakul et al.

Similarly there is some recent work on inspecting the spectral properties of the latent space of latent diffusion models which needs discussion in the main text:

a. Improving the Diffusability of Autoencoders, Skorokhodov et al.

b. EQ-VAE: Equivariance Regularized Latent Space for Improved Generative Image Modeling, Kouzelis et al.

4. **Re. Empirical Results**:

a. The analysis in Section 3.2 is quite interesting. While the t-SNE visualizations are interesting, I think the findings related to the discriminative ability of different feature extractors in Fig. 4a can be made more concrete by demonstrating the top-k accuracy using a classifier head on top of these features extractors.

b. Can the authors provide the loss training curves (somewhere in the appendix maybe) to demonstrate the stability of the SVG training pipeline?

c. Did the authors experiment with fitting diffusion models on the feature space of intermediate layers in the encoder? I understand this could be quite compute intensive for an ablation but could provide valuable insights into what feature spaces are good for working with diffusion models.

d. From Table 4, the advantages of distribution alignment do not seem to statistically significant and therefore unclear. I’m curious why the authors decided to keep this component as a part of their core pipeline?

5. **Minor**: Is there a typo in the heading of Section 3.3 (Visual Feature Generation vs Feature Visual Generation)?

**Questions:**

See weaknesses

---

> ### Author Response · Authors · 2025-11-21
>
> We thank Reviewer go1M for the valuable feedback. We address your concerns and questions below.
>
> >  W denotes Weakness, Q denotes Question.
>
> > ### W1: About inference efficiency
>
> **[Reply]**:
>
> We have revised the phrasing in the abstract for accuracy. It is indeed true that latent diffusion models (LDMs) are inherently more efficient than pixel-space diffusion models due to the lower-dimensional latent representation. Our intended point is that, **even within LDMs, there remains room to improve sampling efficiency**.
>
> Compared to standard VAE-based approaches, SVG achieves **high-quality results with fewer sampling steps**. This improved efficiency is explained by **Figure 4b**: in the SVG latent space, **sampling trajectories under different conditions are more widely separated**, allowing the model to learn the denoising direction more accurately. Consequently, discrete-step approximations introduce smaller errors, enabling fewer step sampling.
>
> Moreover, SVG is **compatible with fast samplers such as DPM solver**. We have conducted additional experiments, further highlighting the practical efficiency of our approach. Results are reported after 80 epochs training without cfg.
>
> | **Model**  | **Sampler** | **Steps** | **gFID↓** |
> | -- | -- | - | - |
> | SiT-XL| Euler  | 5  | 69.38|
> | SiT-XL  | DPM-Solver  | 5  | 41.24|
> | SiT-XL| Euler  | 10| 32.81|
> | SiT-XL  | DPM-Solver  | 10 | 31.87|
> | SiT-XL| Euler  | 50| 20.47|
> | SiT-XL  | DPM-Solver  | 50| 19.24|
> | SiT-XL  | Euler  | 250| 17.20|
> |  |  |  |  |
> | **SVG-XL** | Euler  | 5  | 12.26|
> | **SVG-XL** | DPM-Solver  | 5  | **11.19** |
> | **SVG-XL** | Euler  | 10  | 9.39 |
> | **SVG-XL** | DPM-Solver  | 10 | **9.22**  |
>
>
>
>
> > ### W2: About Fig. 1 presentation.
>
> **[Reply]**:
>
> We thank the reviewer for the suggestion. **In response, we have updated the architecture diagram in Figure.1 with a revised color scheme to better distinguish the different methods.**
>
> Additionally, **we have revised the figure caption to provide more informative descriptions**. Specifically, the updated figure now clarifies the differences among the methods as follows:
>
> - (a) Vanilla VAE-based LDM: the diffusion model is trained on the pretrained VAE latent space.
>
> - (b) Diff. Model Feature Alignment: intermediate features of the diffusion model are aligned to Visual Foundation Model (VFM) features.
> - (c) VAE and Diff. Model Feature Alignment: both VAE latent features and diffusion model intermediate features are aligned to VFM features.
> - (d) Our method: the diffusion model is trained directly in the SVG space derived from self-supervised representations (DINOv3).
>
>
>
> > ### W3: About related work.
>
> **[Reply]**:
>
> We thank the reviewer for raising this important point. Our statement that “VAE latent spaces inherently lack semantic separability” is **based on our empirical observations**:
>
> (1) the t-SNE visualizations in **Figure. 4a** show that class clusters in the VAE latent space are highly entangled, and
>
> (2) linear probing on ImageNet yields **near-random** performance for SDVAE, EQVAE[1], and MARVAE, while even VAVAE exhibits only **marginal separability**.
>
> Our SVG feature space exhibits strong semantic separability, as evidenced by its well-structured t-SNE clusters and near-DINOv3 performance under linear probing. We have clarified this in the revision and explicitly state that our conclusion is grounded in these analyses rather than a general theoretical claim.
>
> Regarding Line 73 in Submission Paper, we have added the missing citation for DINOv3.
>
> We acknowledge that some prior good works have  investigated improving VAE representations [1,2,3,4,5,6], and we have incorporated the suggested related work and discussed its connection to our approach. Our approach to semantic separability builds upon, yet differs from, the disentanglement concept in β-VAE[2]. β-VAE's pioneering contribution is demonstrating that a model can **unsupervisedly** discover and align its latent dimensions with semantic information by enforcing a **strong information bottleneck**. However, this alignment is an **indirect consequence of the training objective**.
> Similarly, EQ-VAE[1] and Diffusability[3] focus on improving the equivariance and diffusability of the VAE latent space, which **indirectly** improve semantic seperability, and works such as[4,5,6] introduce diffusion processes into VAE reconstruction.
>
> `Collectively, these approaches attempt to enhance the semantic discriminability of VAE latent spaces through indirect means rather than directly addressing their underlying semantic structure.` As a result, the improvements remain limited and still **fall far short of** the strong semantic discriminability exhibited by modern visual foundation model (VFM) representations. **In contrast, the feature space used in SVG is derived from a semantically rich backbone, and is no longer constrained by the modeling limitations of the VAE latent space.**

---

> > ### Author Response · Authors · 2025-11-21
> >
> > > ### W4.a: About classification performace of other feature extractors.
> >
> > **[Reply]**:
> >
> > Thank you for the valuable suggestion. To quantify the discriminative ability of the feature extractors shown in **Figure. 4a**, we conducted an additional evaluation by training a simple linear classifier head on top of the extracted features. The top-1 and top-5 accuracy results are summarized as follows. As expected, SDVAE, EQVAE, and MARVAE exhibit **almost no** classification capability, while VAVAE shows **limited** discriminative power. **In contrast, our proposed method demonstrates significantly higher top-k accuracy, confirming its superior feature separability.**
> >
> > | Method          | Top-1 Acc.↑ (%) | Top-5 Acc.↑ (%) |
> > | --------------- | :-------------: | :-------------: |
> > | SDVAE           |      1.27       |      4.74       |
> > | EQVAE           |      1.47       |      5.04       |
> > | MARVAE          |      5.30       |      14.95      |
> > | VAVAE           |      26.02      |      52.30      |
> > | DINOv3-s16plus  |      81.71      |      95.79      |
> > | **SVG Encoder** |      81.80      |      95.87      |
> >
> > We have updated Figure 4a with linear probe results on ImageNet-1K.
> >
> >
> >
> > > ### W4.b: About loss training curves.
> >
> > **[Reply]**:
> >
> > Thank you for the suggestion. We have added the training loss curves (**Figure 14**) to the appendix to illustrate the stability of the SVG training pipeline. The curves show smooth convergence without oscillations or divergence, confirming that our method is stable throughout training. Notably, **all model scales(B,L,XL) demonstrate similarly stable convergence behavior**, indicating that our training pipeline is robust across different model capacities.
> >
> >
> >
> > > ### W4.c: About fitting diffusion models on intermediate layer features
> >
> > **[Reply]**:
> >
> > Thank you for the insightful question. We conduct extensive experiments investigating diffusion models trained on the feature spaces of various intermediate layers in the encoder. These experiments include multiple training runs across different layers, and the supplementary materials provide t-SNE visualizations (**Figure 10**) to illustrate the characteristics of these feature spaces.
> >
> > Our results show that features from **shallower layers** generally lead to **poor generative performance** and weak discriminative capability, despite potentially achieving reasonable reconstruction. This is likely because **shallow features preserve more fine-grained image details, but do not capture semantic information sufficiently**.
> >
> > We hope that these findings clarify your concerns about which encoder feature spaces are most suitable for diffusion-based generation.
> >
> > | Layer | Training Epochs | rFID↓ | gFID↓ |
> > | :---: | :-------------: | :---: | :---: |
> > |   4   |        5        | 0.792 | 73.45 |
> > |   4   |       20        | 0.447 |   -   |
> > |   8   |        5        | 1.309 | 36.63 |
> > |   8   |       20        | 0.859 |   -   |
> > |  11   |        5        | 1.753 | 27.00 |
> > |  11   |       20        | 0.992 |       |
> >
> > ( Training Epochs indicate the number of epochs used to train the SVG Autoencoder. The gFID metrics are obtained after 200K training steps of the SVG-B diffusion model, without classifier-free guidance. )
> >
> >
> >
> >
> > > ### W4.d: About distribution alignment
> >
> > **[Reply]**:
> >
> > We thank the reviewer for the comment. Through extensive experiments, we found that removing the distribution alignment step leads to a substantial drop in generative performance. Specifically, the gFID score increases from **6.11** with distribution alignment to **9.03** without it, indicating a significant degradation in generation quality.
> >
> > We also include t-SNE visualizations of the SVG feature space (with and without distribution alignment) in **Figure 10**, which clearly demonstrate that omitting distribution alignment substantially disrupts the semantic structure of the base DINO space.
> >
> > Therefore, we retain this component as a core part of our pipeline to maintain reliable and high-quality generative performance.
> >
> >
> > > ### W5: About typo.
> >
> > **[Reply]**: Thank you for pointing this out. We have corrected it.
> >
> >
> >
> > *Reference*
> >
> > *[1]. EQ-VAE: Equivariance Regularized Latent Space for Improved Generative Image Modeling, Kouzelis et al.*
> >
> > *[2]. Understanding disentangling in β-VAE, Burgess et al.*
> >
> > *[3]. Improving the Diffusability of Autoencoders, Skorokhodov et al.*
> >
> > *[4]. Score-based Generative Modeling in Latent Space, Vahdat et al.*
> >
> > *[5]. DiffuseVAE: Efficient, Controllable and High-Fidelity Generation from Low-Dimensional Latents, Pandey et al.*
> >
> > *[6]. Diffusion Autoencoders: Toward a Meaningful and Decodable Representation, Preechakul et al.*

---

### Official Review · Reviewer_ANZt · 2025-11-01

**Soundness:** 2
**Presentation:** 3
**Contribution:** 4
**Rating:** 6
**Confidence:** 3

**Summary:**

- The authors claim that the latent space of VAE is not generation-friendly.  Instead of VAE, they propose a self-supervised representation for Visual Generative (SVG).
- SVG autoencoder maintains the dimension of the DINO feature and augments the residual network to enhance reconstruction.
- While training diffusion models in high-dimensional spaces is generally challenging, the well-dispersed semantic structure of SVG features makes training stable and efficient.

**Strengths:**

- They first introduce to directly employ the dimension of DINO v3 feature and show that the well-dispersed semantic feature enables training in a high-dimensional space
- The feature visualization in Fig. 4 and toy experiments support the author's hypothesis that the latent space of VAE is not generation-friendly.

**Weaknesses:**

- Lack of explanation of architectural choice (Residual encoder)

**Questions:**

- Have you tried other architectures except for the Residual encoder, such as adapting LoRA to DINO, or employing other foundational vision encoders? The explanation will be helpful to strengthen the paper.

---

> ### Author Response · Authors · 2025-11-21
>
> We thank Reviewer ANZt for the valuable feedback. We address your concerns and questions below.
>
> >  W denotes Weakness, Q denotes Question.
>
> > ### Q1: About other VFM choice.
>
> **[Reply]**:
>
> Thank you for the question. In **Table 3**, we compare several encoder choices, including the MAE encoder, SigLIP2, DINOv2, and DINOv3. Our results show that SigLIP2 provides unsatisfactory reconstruction quality, and both SigLIP2 and the MAE encoder are less suitable for downstream perception tasks [1,2] Based on these observations, we adopt DINOv3 as our primary backbone.
>
> *Reference*
>
>  *[1] DINOv2: Learning Robust Visual Features without Supervision,  arXiv:2304.07193*
>
>  *[2] DINOv3, arXiv:2508.10104*
>
>
>
> > ### W1/Q1: About residual encoder design.
>
> **[Reply]**:
>
> We agree that clarifying our architectural choice for the residual encoder is important. Our design philosophy is to keep the framework **minimal and general** while ensuring that the semantic representation provided by DINOv3 is complemented with the fine-scale information needed for accurate reconstruction.
>
> To further assess sensitivity and rationale of the residual design, we conducted two additional studies: (1) replacing the residual module with a Swin Transformer and (2) applying LoRA fine-tuning to the frozen DINOv3 encoder. The results show that **LoRA improves tokenization quality (lower rFID) but significantly harms generative quality (higher gFID), while changing the residual backbone slightly improves reconstruction but degrades generation**, offering no clear advantage. For the table below, rFID is measured after training the Tokenizer for 5 epochs, and gFID is measured after training the diffusion model for 40 epochs.
>
>
> | **Method**        | rFID↓ | gFID↓ |
> | :---------------- | :---: | :---: |
> | DINOv3+LoRA       | 1.158 | 89.97 |
> | Residual (Swin-T) | 1.676 | 29.08 |
> | Residual (ViT-S)  | 1.753 | 27.00 |
>
> As illustrated in Figure 5, DINOv3 already provides sufficient semantic structure to recover the global layout, and the residual branch primarily contributes color refinement and high-frequency detail resotration. Because its role is restricted to such fine-grained adjustments, the overall system is **not highly sensitive** to the architectural specifics of the residual encoder.

---

### Official Review · Reviewer_w1GX · 2025-11-04

**Soundness:** 3
**Presentation:** 3
**Contribution:** 3
**Rating:** 8
**Confidence:** 4

**Summary:**

This paper proposes replacing the VAE-based latent space commonly used in latent diffusion models with a latent space derived directly from a pretrained self-supervised vision model, specifically DINOv3. The authors argue that the DINOv3 feature space already encodes strong semantic structure, making it a more suitable domain for diffusion than the low-level reconstruction-oriented latent space produced by VAEs. To enable high-fidelity image reconstruction from this feature space, the method introduces a lightweight residual decoder that is trained to recover fine-grained visual details. A diffusion model is then trained directly in this semantic latent space.

The paper provides extensive experiments demonstrating improved generation quality and training efficiency compared to standard VAE-based latent diffusion baselines. The authors also show that the resulting latent space preserves semantic separability and can benefit downstream recognition or manipulation tasks.

**Strengths:**

1. Novel and intuitive idea. While prior work has attempted to align VAE latent spaces with features from vision foundation models to improve semantic consistency, this paper takes a more direct and conceptually clean approach by completely replacing the VAE encoder with pretrained DINOv3 features. This eliminates the need to engineer semantic structure into the latent space and leverages an already well-organized representation.
2. Comprehensive empirical validation. The authors present an extensive set of experiments, covering comparisons with multiple baselines, ablation studies, visual quality assessments, and evaluation across different datasets. The breadth of experiments strengthens the credibility of the proposed approach.
3. Strong performance gains. The method achieves consistently improved generation quality while also demonstrating benefits in efficiency and semantic controllability.

**Weaknesses:**

1. Limited analysis of the residual decoder design. While the proposed residual module plays a critical role in reconstructing high-frequency details, the paper does not provide sufficient ablations on its architecture or training strategy. In particular, it is unclear how sensitive the overall performance is to this component, and the paper does not report results using only the DINOv3 features without the residual correction branch. This makes it difficult to assess how much of the performance gain comes from the semantic latent space itself versus the added decoder capacity.

**Questions:**

1. Classifier-free guidance behavior. The paper mentions that classifier-free guidance has a reduced effect in this semantic latent space, but no dedicated experiment or quantitative analysis is provided. Could the authors clarify why guidance becomes less influential, and provide empirical evidence illustrating how guidance scales differ compared to standard VAE-based latent diffusion?
2. Interpolation behavior in the semantic latent space. Although the latent representation is said to be semantically well-structured, the interpolation results sometimes show abrupt semantic changes rather than smooth transitions. How do the authors interpret this phenomenon? Does it indicate that the latent space is organized more in a clustered manner rather than forming smooth semantic manifolds?
3. Extension to video generative models. Do the authors believe that this architecture can be extended to video (i.e., replacing video VAEs with video or image foundation model features)? If so, what modifications would be necessary to handle temporal consistency and motion priors? Any discussion on temporal latent alignment would be valuable.
4. Model size and decoding efficiency. Compared to a standard VAE encoder–decoder pipeline, how does using a vision foundation model affect computational overhead and inference latency? In particular:
What is the relative model size change?
Does the residual decoder introduce additional decoding cost?
Is end-to-end sampling faster or slower in practice?

---

> ### Author Response · Authors · 2025-11-21
>
> We sincerely thank Reviewer w1GX for the valuable insights and constructive feedback. Below, we address your concerns and questions in detail.
>
> > W denotes Weakness, Q denotes Question.
>
> > ### W1: About SVG Autoencoder design.
>
> **[Reply]**:
>
> Thank you for raising this point. `Our design philosophy is to develop a general and minimal framework for replacing the VAE latent space with a semantically structured representation`. In line with this principle, the residual branch is introduced **only on the encoder side** to supplement the DINOv3 features with the fine-scale color variations and high-frequency details that DINOv3 does not explicitly model. The decoder follows the VAVAE-Decoder architecture, with only the input channel dimension modified; its capacity is therefore unchanged relative to the baseline.
>
> Our residual branch of the encoder is deliberately lightweight: we use a ViT-S (from timm) as the residual backbone, and only the residual branch and decoder are updated, supervised by L1, LPIPS, and GAN losses.
>
> ###### 1. Abaltion of Different Encoder Design
>
> To further assess sensitivity to the residual design, we conducted two additional studies: (1) replacing the residual module with a Swin Transformer-Tiny and (2) applying LoRA fine-tuning to the frozen DINOv3 encoder. The results show that **LoRA improves tokenization quality (lower rFID) but significantly harms generative quality (higher gFID), while changing the residual backbone slightly improves reconstruction but degrades generation**, offering no clear advantage. For the table below, rFID is measured after training the Tokenizer for 5 epochs, and gFID is measured after training the SVG-B for 40 epochs.
>
>
> | **Method**        | rFID↓ | gFID↓ |
> | :---------------- | :---: | :---: |
> | DINOv3+LoRA       | 1.158 | 89.97 |
> | Residual (Swin-T) | 1.676 | 29.08 |
> | Residual (ViT-S)  | 1.753 | 27.00 |
>
> As illustrated in **Figure 5**, DINOv3 already provides sufficient semantic structure to recover the global layout, and the residual branch primarily contributes color refinement and high-frequency detail resotration. Because its role is restricted to such fine-grained adjustments, the overall system is **not highly sensitive** to the architectural specifics of the residual encoder.
>
> ###### 2. Effectiveness of SVG Autoencoder Design
>
> Finally, the comparison between using **DINOv3 features alone** and **the full SVG design** is reported in the original paper (**Table 4**), and the results are shown below. Reconstruction performance is reported after 40 epochs of training, and gFID is measured after training the diffusion XL size model for 100 epochs using classifier-free guidance.
>
> | **Method** |  rFID↓   |  gFID↓   |
> | :--------- | :------: | :------: |
> | DINOv3     |   1.17   |   6.12   |
> | **SVG**    | **0.65** | **6.11** |
>
> While **DINOv3 alone already yields competitive generative quality, its reconstruction performance is considerably weaker**. This limitation directly affects downstream applications such as editing, where accurate recovery of colors and high-frequency details is essential. Taken together, these observations justify the necessity and rationality of our residual encoder design.
>
>
>
>
> > ### Q1: About classifier-free guidance behavior.
>
> **[Reply]**:
>
> Thank you for the question. We have added a dedicated comparison showing how performance varies with different classifier-free guidance (CFG) scales in **Figure 13**. `The results confirm that CFG still improves generation quality in our method.`
>
> Specifically, **our method already achieves strong performance with CFG=1 (i.e., without guidance), substantially outperforming the corresponding baseline in the VAE+Diffusion setting. Applying CFG further enhances results, but the overall gain from guidance is less pronounced compared to VAE-based models**.
>
> We hypothesize that this behavior arises from the same phenomenon illustrated in **Figure 4b**. In standard VAE+diffusion paradigms, conditional outputs are often inaccurate, so CFG relies heavily on the unconditional prediction to correct the denoising direction. In contrast, in the SVG+Diffusion framework, the conditional outputs are already relatively accurate, meaning that guidance provides only a moderate additional improvement, while the model’s baseline performance is already strong.

---

> > ### Author Response · Authors · 2025-11-21
> >
> > > ### Q2: About interpolation behavior.
> >
> > **[Reply]**:
> >
> > We observe that interpolations in the semantic latent space are generally continuous. In Figures 8 and 9 (third row), the apparent abrupt change in the bird’s orientation in the SVG results likely stems from the coarse-grained nature of the semantic conditioning in class-to-image tasks.
> >
> > To further clarify the structure of the latent space, we provide additional t-SNE visualizations of the SVG feature space in **Figure 10**. These show that features from the same semantic category form tight clusters, while different categories are well-separated. This suggests that the latent space is organized in a semantically meaningful, clustered manner rather than forming perfectly smooth manifolds across all categories.
> >
> >
> > > ### Q3: About extension to video generation.
> >
> > **[Reply]**:
> >
> > **We believe that the core idea of SVG can be naturally extended to video generative models.** To ensure temporal consistency and incorporate motion priors, temporal convolutions or temporal attention modules could be integrated into the autoencoder structure.
> >
> > As a preliminary exploration, we extracted DINOv3 features for each frame of continuous video and observed that these features remain **temporally coherent**. Moreover, DINOv3 features already support tasks such as segmentation, demonstrating that they capture meaningful structural information across frames. Based on these observations, applying the SVG methodology to video reconstruction and generation is expected to facilitate the synthesis of complex, structured motion while maintaining semantic consistency over time.
> >
> >
> >
> > > ### Q4: About Tokenizer size and decoding efficiency.
> >
> > **[Reply]**:
> >
> > Thank you for the question. We have analyzed the model size, computational cost, and inference latency of the SVG autoencoder in comparison with standard VAE-based pipelines such as SD-VAE and VA-VAE.
> >
> > | **Module**      | #Params | GFLOPs | Latency(ms/img) | Throughput(imgs/s) |
> > | :-------------- | :-----: | :----: | :-------------: | :----------------: |
> > | SDVAE-Encoder   | 34.16 M | 135.59 |      7.54       |       224.22       |
> > | VAVAE-Encoder   | 28.41 M | 69.21  |      6.45       |       332.87       |
> > | DINOv3-s16plus  | 28.70 M |  7.48  |      11.49      |      2118.88       |
> > | **SVG-Encoder** | 39.89 M | 10.33  |      14.28      |      1555.07       |
> > |                 |         |        |                 |                    |
> > | SDVAE-Decoder   | 49.49 M | 310.62 |      13.76      |       114.26       |
> > | VAVAE-Decoder   | 41.42 M | 126.56 |      9.43       |       198.60       |
> > | **SVG-Decoder** | 43.08 M | 126.94 |      9.33       |       199.68       |
> > |                 |         |        |                 |                    |
> > | SiT-XL          |    -    |   -    |     426.48      |       10.56        |
> >
> > In our evaluation, throughput measures the average processing speed under a large batch size (64) setting, while latency reports the end-to-end inference time for single-image inference (i.e. batch size = 1).
> >
> > The results show that the SVG encoder introduces only a modest increase in parameters while requiring even lower computational cost. The residual decoder contributes negligible overhead, and end-to-end sampling with the SVG autoencoder remains comparable in speed to the baseline methods. Notably, the **encoder FLOPs of SVG are significantly lower** than other methods, while its **throughput is substantially higher**, indicating potential advantages in large-scale computation scenarios.

---

### Official Review · Reviewer_j2Au · 2025-11-08

**Soundness:** 3
**Presentation:** 3
**Contribution:** 3
**Rating:** 6
**Confidence:** 4

**Summary:**

The paper proposes SVG, a latent diffusion framework that removes the VAE encoder and instead uses frozen DINOv3 features as the semantic backbone, augmented with a lightweight residual encoder for fine details. A decoder maps the concatenated features back to pixels; diffusion is trained directly in this high-dimensional feature space using a flow-matching setup. The authors argue this yields a latent space with better semantic separability, enabling faster convergence and competitive few-step sampling on ImageNet-256. Key components include (i) the DINO-based encoder + residual branch, (ii) a distribution alignment step for the residual features, and (iii) a standard DiT/SiT-style diffusion trained over the new latent space. Reported results show improved few-step FID/IS and faster training vs. VAE-based baselines, with additional qualitative analyses.

**Strengths:**

- Replaces VAE latents with a semantically structured feature space (frozen DINOv3) and adds a Residual Encoder to recover high-frequency detail for reconstruction. The training pipeline is simple and reuses standard diffusion tooling.

- At 80 epochs / 25 steps, SVG-XL reports good generation performance and improves further with more training, indicating strong few-step behavior relative to baselines.

- The velocity toy study ties semantic dispersion to optimization ease and step efficiency; paired with t-SNE, it provides an intuitive explanation for why VLM feature space diffusion may converge faster.

**Weaknesses:**

- Missing REPA-XL 80-epoch (CFG) numbers.

Table 1 lists REPA-XL (80 epochs, 250 steps) only without CFG and with missing IS, whereas the 800-epoch row provides both CFG and no-CFG. Since the central claim is superior data/compute efficiency, the 80-epoch CFG result for REPA-XL is essential for a fair comparison.

- Tokenizer model-size comparison not shown in Table 1.

The system table includes #params for the generation model but omits tokenizer sizes. Given SVG uses frozen DINOv3 + Residual Encoder, it'd be helpful to have a side-by-side tokenizer capacity comparison (params and MACs) against SD-VAE / VA-VAE to contextualize efficiency and scaling claims.

- Unclear scaling to higher resolution.

The Limitations section explicitly states that higher resolutions and larger datasets remain “underexplored.” Given VLM feature space diffusion relies heavily on the base frozen VLM, the method may inherit DINOv3’s resolution constraints. Please clarify whether scaling requires a different backbone or architectural changes. Empirical evidence beyond 256×256 and ideally 1024+ resolution is very helpful to justify the potential for real-world applications such as T2I.


- Overclaims in the Introduction.

The paper claims to “fully retain DINOv3’s strengths beyond generation”, but the addition of a Residual Encoder and decoder training could alter the representational geometry. This needs quantitative evidence (e.g., linear probing / retrieval / segmentation probes done in the combined space, not just DINO).

It also states prior alignment approaches are “ad hoc fixes” that “do not fundamentally alter … latent space structure.” This contradicts the improved results of the baselines mentioned here. In my opinion, these methods absolutely alter the latent space structure.


- Insufficient detail on residual-distribution alignment.

Section 3.2 notes: “we align the Residual Encoder outputs with the DINO feature distribution” but does not specify how. Sensitivity to alignment strength and its effect on semantic separability vs. reconstruction is crucial; please add technical details and an ablation sweep.

- Semantic analysis of the new latent space is limited.
t-SNE and the velocity toy example are helpful, but they don’t isolate the effect of the residual branch on semantics. Provide representation probes on the final concatenated space (not just DINO alone): e.g., linear probing, can be helpful.

**Questions:**

- REPA-XL (80-epoch, CFG): Can you report CFG FID/IS for REPA-XL at 80 epochs in Table 1 to support efficiency claims under equalized compute?

- Tokenizer size & cost: Please provide tokenizer parameter counts and FLOPs/MACs (encoder+decoder) for SD-VAE / VA-VAE / SVG in a single table to make system-level comparisons fair.

- High-resolution scaling: What changes (if any) are needed to scale SVG to 512/1024? Any preliminary 512×512 results?

- Alignment details: How exactly is the residual aligned to DINO (loss form, feature layers tapped, λ values)? Sensitivity plots would help.

- Semantic probes on the final space: Can you add linear-probe / t-SNE on the concatenated SVG features (with and without alignment) to substantiate the “retains DINO strengths” claim?

---

> ### Author Response · Authors · 2025-11-21
>
> We thank Reviewer j2Au for the positive feedback. We address the concerns as described below.
>
> > W denotes Weakness, Q denotes Question.
>
> > ### W1/Q1: About missing numbers in Table 1.
> >
>
> **[Reply]**:
>
> Thank you for pointing this out. We agree that including the 80-epoch (with CFG) REPA-XL results leads to a more comprehensive and fair comparison.
>
> These numbers were not reported in the original submission **because the REPA paper (arXiv:2410.06940) does not provide them, and the official GitHub repository does not release the corresponding pretrained checkpoints.** To address this gap, we retrained REPA-XL following the official implementation and report both gFID and IS (with and without classifier-free guidance in Table 1).
>
> ###### Clarification of Our Cenetral Claim
>
> We also appreciate the opportunity to clarify our central claim. REPA is an excellent method for optimizing and aligning representations in the VAE latent space. **However, our goal is fundamentally different: we aim to construct a unified feature space for generation, and to demonstrate its strong perceptual capabilities.** To this end, we take a more aggressive approach by fully discarding the VAE latent space.
>
> While SVG-XL surpasses REPA-XL at 80 epochs using **25× fewer sampling steps** *without* classifier-free guidance, REPA-XL achieves lower FID *with* classifier-free guidance. Following REPA strategy, we also align the layer-8 hidden states with the DINO encoder for early-layer representation alignment (we call it ELRA), and we find that this technique also improves SVG’s performance. However, because this method operate within the same feature space, ELRA is less effective in our setting.
>
> **Takeways:** Our core objective is to demonstrate that a task-unified feature space has strong potential for unified visual understanding, perception, and generation. By fully abandoning the VAE-space and adopting a more aggressive design, SVG still highlight the effective and efficient diffusion training and sampling.
>
>
>
> | **Model**        | Epoch | Sampling Steps | FID50K w/o CFG | FID50K w/ CFG |
> | :--------------- | :---: | :------------: | :------------: | :-----------: |
> | SiT-XL           |  80   |       10       |     32.81      |     10.26     |
> | SiT-XL           |  80   |       25       |     22.58      |     6.06      |
> | SiT-XL           |  80   |      250       |     17.20      |     5.10      |
> | REPA-XL          |  80   |       10       |     22.21      |     5.80      |
> | REPA-XL          |  80   |       25       |     12.13      |     3.01      |
> | REPA-XL          |  80   |      250       |      7.90      |     2.65      |
> |                  |       |                |                |               |
> | SVG-XL (w/ REPA) |  80   |       10       |      6.37      |     3.61      |
> | SVG-XL           |  80   |       25       |      6.57      |     3.54      |
> | SVG-XL (w/ REPA) |  80   |       25       |      5.82      |     3.35      |
>
>
>
>
>
> > ### W2/Q2: About Tokenizer model-size comparison.
>
> **[Reply]**:
>
> Thank you for the suggestion. We have added the parameter counts and computational cost of the tokenizer in the revised version (Table 7). Our results show that the **SVG Autoencoder has a comparable number of parameters and overall latency to the baselines**, with differences in latency being negligible relative to the backbone.
>
> Notably, the **encoder FLOPs of SVG are significantly lower** than other methods, while its **throughput is substantially higher**, indicating potential advantages in large-scale computation scenarios.
>
> | **Module**      | #Params | GFLOPs | Latency(ms/image) | Throughput(image/s) |
> | :-------------- | :-----: | :----: | :---------------: | :-----------------: |
> | SDVAE-Encoder   | 34.16 M | 135.59 |       7.54        |       224.22        |
> | VAVAE-Encoder   | 28.41 M | 69.21  |       6.45        |       332.87        |
> | DINOv3-s16plus  | 28.70 M |  7.48  |       11.49       |       2118.88       |
> | **SVG-Encoder** | 39.89 M | 10.33  |       14.28       |       1555.07       |
> |                 |         |        |                   |                     |
> | SDVAE-Decoder   | 49.49 M | 310.62 |       13.76       |       114.26        |
> | VAVAE-Decoder   | 41.42 M | 126.56 |       9.43        |       198.60        |
> | **SVG-Decoder** | 43.08 M | 126.94 |       9.33        |       199.68        |
> |                 |         |        |                   |                     |
> | SiT-XL          |    -    |   -    |      426.48       |        10.56        |
>
> In our evaluation, throughput measures the average processing speed under a large batch size (64) setting, while latency reports the end-to-end inference time for single-image inference (i.e. batch size = 1).

---

> > ### Author Response · Authors · 2025-11-21
> >
> > > ### W1/Q3: About scaling to higher resolution.
> >
> > **[Reply]**:
> >
> > We acknowledge that assessing the model’s behavior at higher resolutions is important for evaluating its practical effectiveness. Although our main experiments use 256×256, the original DINOv3 paper (arXiv: 2508.10104) explicitly states that the backbone supports a broad and various input-resolution range, including 512×512 and 1024×1024, which we have also independently verified.
> >
> > To further verify this capability, we conducted additional reconstruction and generation experiments at 512×512 and 1024×1024. For the SVG Autoencoder, we first resumed from the checkpoint trained at 256×256 for 40 epochs, then continued training at 512×512 for 5 epochs, and finally performed one epoch of finetuning at 1024×1024. For the diffusion model, we resumed from the SVG-XL checkpoint trained at 256×256 for 300 epochs, subsequently finetuned it at 512×512 for 25 epochs, and completed an additional epoch at 1024×1024. **The corresponding visualizations are included in the appendix (Figure 15-18), and confirm that our method scales to higher resolutions without any architectural modifications**.
> >
> >
> >
> > > ### W4/W6/Q5: About semantic probes on the SVG space.
> >
> > **[Reply]**:
> >
> > We fully acknowledge the concern regarding whether the representational geometry is preserved after introducing the Residual Encoder. Our analysis provides strong empirical evidence that **the SVG feature space faithfully maintains both the geometric structure and the downstream utility of the original DINOv3 representations.**
> >
> > First, Table 4 in the original submission already includes a direct quantitative comparison between the original DINOv3 features (first row) and our SVG combined space (third row) across image classification, semantic segmentation, and depth estimation. **SVG consistently matches the performance of DINOv3**, demonstrating that downstream perception capability is reliably retained.
> >
> > | Tokenizer | ImageNet-1k Top-1↑ | ADE20K mIoU↑ | NYUv2 RMSE↓ |
> > |-|-|-|-|
> > |DINOv3|81.71| 46.37 | 0.362 |
> > |**SVG**|**81.80**  | **46.51** | **0.361**   |
> >
> > In addition, we have added t-SNE visualizations of DINOv3 and SVG (w/ and wo distribution alignment) space in the appendix to assess structural preservation. **Figure 10** further validates that our design **does not distort the representational structure learned by DINOv3**.
> >
> > Collectively, these results substantiate our conclusion that SVG preserves both the representational geometry and the downstream effectiveness of DINOv3.
> >
> >
> >
> > > ### W4: About latent space structure of baseline models.
> >
> > **[Reply]**:
> >
> > We appreciate the reviewer’s point and have revised the statements for precision. Our intention was not to suggest that prior alignment methods leave the latent space unchanged. Rather, while these baselines do introduce modifications within the original VAE latent space, **the resulting semantic structure remains limited by the representational capacity of VAE space**. This can be observed in the linear-probe results on ImageNet-1K and t-SNE plots (**Figure 4a**).
> >
> > | Method | Top-1 Acc.↑ (%) | Top-5 Acc.↑ (%) |
> > |-|:-:|:-:|
> > |SDVAE|1.27|4.74|
> > |EQVAE|1.47|5.04|
> > |MARVAE|5.30|14.95|
> > |VAVAE|26.02|52.30|
> > |DINOv3-s16plus|81.7|95.79|
> > |**SVG Encoder**|81.80|95.8|
> >
> > In contrast, **our approach replaces the low-dimensional VAE latent space entirely with the high-dimensional DINOv3 feature space, which is intrinsically endowed with strong semantic organization**. This represents a substantially more fundamental change than the adjustments made by previous baselines. Moreover, our method constructs a latent space that is **task-agnostic and generalizable across downstream tasks**, a capability that previous alignment apporaches cannot achieve.
> >
> > We hope this resolves the perceived contradiction and more accurately reflects the conceptual difference between prior work and our SVG design.
> >
> >
> >
> > > ### W5/Q4: About distribution alignment details.
> >
> > **[Reply]**: We have clarified **Section 3.3** regarding the residual–DINO feature alignment. Our method is deliberately simple and requires no hyperparameter tuning: for each training batch, we compute the mean and standard deviation of the DINOv3 features and directly normalize the residual features to match these statistics.
> > In specific, for each training batch, let $F_D$ denote the DINOv3 features and $F_R$ the residual features. The alignment is performed by normalizing $F_R$ to match the batch statistics of $F_D$:
> > $$
> > \hat{F}_R = \frac{F_R - \mu(F_R)}{\sigma(F_R)} \cdot \sigma(F_D) + \mu(F_D)
> > $$
> >
> > where $\mu(\cdot)$ and $\sigma(\cdot)$ compute the mean and standard deviation along the feature dimension. This ensures that the residual features are aligned to the base DINO feature distribution.
> >
> > **This straightforward approach avoids sensitivity issues** while reliably preserving the semantic geometry of the base DINO space, as evidenced by the strong reconstruction and generation results reported in **Table 4**.

---

### Author Response · Authors · 2025-11-30
**General Response to Area Chair and Reviewers**

Dear Area Chair and Reviewers,

We sincerely appreciate the time and effort you have dedicated to reviewing our manuscript.

We introduce **SVG**—a novel latent diffusion model that replaces the traditional VAE with a self-supervised representation autoencoder, leveraging **frozen  pre-trained visual representations** (DINOv3 features in our paper) augmented with a **lightweight residual branch** for efficient and high-fidelity visual generation. This approach creates a semantically structured latent space that enables accelerated training and efficient sampling. As the reviewers highlighted, the paper presents a **novel and intuitive** idea (w1GX, ANZt), introduces **simple and clean design choices** (ANZt, go1M), and is **supported by strong empirical results**, demonstrating improved generation quality and efficiency (j2Au, w1GX).

We greatly appreciate your constructive feedback. In response, we have carefully revised and enhanced the manuscript by providing extensive additional analysis and experiments:

- **Comprehensive Efficiency Comparison** (Table 7, Response to j2Au, w1GX): We added a detailed breakdown of parameter count, GFLOPs, latency, and throughput for the full SVG autoencoder system versus VAE baselines, confirming SVG’s efficiency advantages, particularly in encoder throughput.
- **High-Resolution Scalability** (Fig. 15-18, Response to j2Au): We conducted and visualized additional experiments to verify that SVG scales effectively to higher resolutions (up to $1024 \times 1024$) without architectural modifications.
- **Ablation and Rationale of SVG Autoencoder Design** (Response to w1GX, ANZt): We provided further ablation studies (e.g., using Swin-T residual, DINOv3 with LoRA) to justify the minimal and effective design of the residual encoder. We also clarified that the SVG latent space achieves strong reconstruction fidelity while preserving the downstream utility of DINOv3 features (Table 4).
- **Classifier-Free Guidance (CFG) Analysis** (Fig. 13, Response to w1GX): We added a dedicated quantitative analysis demonstrating that while CFG remains beneficial, its effect is modest in SVG due to the already accurate conditional outputs facilitated by the semantic latent space.
- **Distribution Alignment Details** (Section 3.3, Response to j2Au): We provided explicit technical details on the simple batch-level feature alignment mechanism, confirming that it requires no hyperparameter tuning, avoids sensitivity issues, and is effective in practice.
- **Missing Baseline Data** (Table 1, Response to j2Au): We retrained and reported the essential REPA-XL 80-epoch (with CFG) results, ensuring a comprehensive and fair comparison to validate our efficiency claims.
- **Semantic Structure and Interpolation** (Fig. 10, Response to j2Au, w1GX): We added t-SNE visualizations of the SVG feature space, confirming that it preserves the clustered, semantically well-separated structure of DINOv3, which aids stability.

In the revised manuscript, these updates are temporarily highlighted in **blue** for your convenience.

We sincerely hope these updates help better convey the benefits of the proposed SVG to the ICLR community. We look forward to further discussion.

Thank you very much,

Authors.

---

### Meta-Review · Area_Chair_5pML · 2025-12-15

**Summary:**

The submission proposes training a latent diffusion model on the latent space of a DINO encoder. It received unanimous positive reviews. Reviewers praised the strong performance, simple and intuitive approach, and extensive experiments. There were minor concerns about a) missing some baseline configurations, b) missing model details, c) lacking demonstration of preserved latent structure. The rebuttal was strong with numerous extra experiments, analysis, and clarifications. I recommend acceptance as a poster.

**Reviewer Concerns:**

The rebuttal was strong with numerous extra experiments, analysis, and clarifications. It seems to cover to every reviewer concern. However, the baseline configurations with CFG raised by j2Au actually perform better than the proposed method, and only qualitative results were provided at higher resolutions. These are not severe enough to prevent acceptance, though.

**Reviewer Scores:**

Scores were already high so I don't believe there would be any changes.

---

### Decision · Program_Chairs · 2026-01-26

Accept (Poster)